# ASκDAGGER: Active Skill-level Data Aggregation for Interactive Imitation Learning

**Jelle Luijkx**                                                          *j.d.luijkx@tudelft.nl*
*Department of Cognitive Robotics*
*Delft University of Technology*

**Zlatan Ajanović**                                          *zlatan.ajanovic@ml.rwth-aachen.de*
*Department of Computer Science*
*RWTH Aachen University*

**Laura Ferranti**                                                      *l.ferranti@tudelft.nl*
*Department of Cognitive Robotics*
*Delft University of Technology*

**Jens Kober**                                                          *j.kober@tudelft.nl*
*Department of Cognitive Robotics*
*Delft University of Technology*

**Reviewed on OpenReview:** *https://openreview.net/forum?id=987Az9f8fT*

## Abstract

Human teaching effort is a significant bottleneck for the broader applicability of interactive imitation learning. To reduce the number of required queries, existing methods employ active learning to query the human teacher only in uncertain, risky, or novel situations. However, during these queries, the novice's planned actions are not utilized despite containing valuable information, such as the novice's capabilities, as well as corresponding uncertainty levels. To this end, we allow the novice to say: "*I plan to do this, but I am uncertain.*" We introduce the Active Skill-level Data Aggregation (ASκDAGGER) framework, which leverages teacher feedback on the novice plan in three key ways: (1) S-Aware Gating (SAG): Adjusts the gating threshold to track sensitivity, specificity, or a minimum success rate; (2) Foresight Interactive Experience Replay (FIER), which recasts valid and relabeled novice action plans into demonstrations; and (3) Prioritized Interactive Experience Replay (PIER), which prioritizes replay based on uncertainty, novice success, and demonstration age. Together, these components balance query frequency with failure incidence, reduce the number of required demonstration annotations, improve generalization, and speed up adaptation to changing domains. We validate the effectiveness of ASκDAGGER through language-conditioned manipulation tasks in both simulation and real-world environments. Code, data, and videos are available at `https://askdagger.github.io`.

## 1 Introduction

The promise of imitation learning is to enable individuals to teach robots to perform desired tasks, all without the need for specialized knowledge in coding or robotics. This learning paradigm is beneficial when humans possess the knowledge to solve a task but prefer not to do it themselves due to its repetitive, risky nature, or when automation is more efficient. Specifically, demonstrating correct robot behavior in unstructured environments can be simpler than engineering a controller. Imitation learning has demonstrated success across various domains, such as autonomous driving (Pomerleau, 1988), helicopter aerobatics (Abbeel et al., 2010), language-conditioned robotic manipulation (Jang et al., 2022; Shridhar et al., 2021), and generalist

robot policies (Brohan et al., 2022; Reed et al., 2022; Octo Model Team et al., 2024). Despite these successes of *behavioral cloning* (BC), it can suffer from *covariate shift.* This issue arises when imitation learning is naively posed as a standard supervised learning problem. In imitation learning, the data is not independent and identically distributed because past predictions can influence future states (Ross et al., 2011). As a result, a prediction error can lead to encountering states unseen in the training data, causing a cascade of mistakes since errors in these unfamiliar states are even more likely.

Covariate shift issues can be alleviated with Interactive Imitation Learning (IIL) methods (Celemin et al., 2022). These methods involve obtaining human demonstrations, corrections, and reinforcements interactively. In a seminal work, Ross et al. (2011) introduced the Dataset Aggregation (DAgger) algorithm. This approach alleviates the covariate shift problem by aggregating human input while executing the novice policy. This enables the novice to learn to recover from failures, for instance. While having favorable performance guarantees, the DAgger algorithm requires continuous teacher input and can have safety issues as the novice policy is executed while learning to perform the task.

To overcome these limitations of the DAgger algorithm, various extensions allow the novice to *actively* query the teacher in risky (Hoque et al., 2022) or uncertain situations (Menda et al., 2019; 2017; Zhang & Cho, 2017; Hoque et al., 2023). We refer to these data aggregation methods as active DAgger approaches, as they integrate data aggregation with active learning. The benefit of this active learning strategy is twofold. First, possible failures can be prevented during the interactive training phase since uncertainty is assumed to be correlated with failures. Second, this strategy minimizes the number of demonstrations needed by maximizing their meaningfulness. That is to say, it is a waste of resources if humans demonstrate behaviors already mastered by novices. Instead, the teacher should only demonstrate what the robot novice can not do to enable learning from as few demonstrations as possible.

Existing active DAgger methods hand over control when querying the human teacher. Instead, we allow the novice to also communicate their planned action when they are uncertain. This allows the teacher to validate or correct the novice plan, providing valuable feedback that can be leveraged in several ways. First, it reveals the levels of uncertainty where the novice succeeds or fails. This information can be used to improve gating by allowing dynamic threshold adjustments to maintain a desired sensitivity, specificity, or minimum system success rate. This extends existing methods, which either require constant supervision (Kelly et al., 2019), rely on heuristics (Zhang & Cho, 2017), or use a fixed query rate independent of novice performance (Hoque et al., 2022). Second, validated novice actions can be aggregated into the demonstration dataset, reducing the need for teacher demonstrations. Additionally, invalid plans may be useful demonstrations for alternative goals if the teacher relabels them accordingly. Third, since not all demonstrations are created equally, we can prioritize replay by considering the validity of the novice plan, the corresponding uncertainty level, and the demonstration age. For example, the novice might learn more from a recent demonstration in a situation where they failed rather than one where they acted successfully.

To this end, we introduce the Active Skill-level Data Aggregation (ASkDAgger) framework, a novel IIL method where the robot novice actively communicates its planned actions when uncertain. An overview of ASkDAgger is shown in Fig. 1, and it is built on three key contributions: i) S-Aware Gating (SAG): Adjusts the gating threshold to maintain a user-specified metric — sensitivity (true positive rate), specificity (true negative rate), or minimum system success rate. ii) Foresight Interactive Experience Replay (FIER): aggregates valid and relabeled novice action plans into the demonstration dataset. iii) Prioritized Interactive Experience Replay (PIER): prioritizes replay based on uncertainty, novice success, and demonstration age.

Since ASkDAgger relies on the novice communicating its planned actions for teacher feedback, the method is most practical for moderate feedback frequencies. ASkDAgger therefore targets mid- to high-level control tasks rather than end-to-end policy learning. It is most applicable in scenarios where a robot has access to predefined parameterizable skills such as grasping, walking, pushing, door opening, screwing, or inserting. In such cases, the robot novice needs to learn the parameters and affordances of these skills given a user-specified command. When querying the teacher, the robot novice can specify which skill they plan to use, along with the parameterization of that skill. If the teacher deems the novice's plan invalid, they can provide a demonstration by annotating the appropriate skill and its parameters. For example, a pick skill can be parameterized by a Cartesian pick position and orientation.

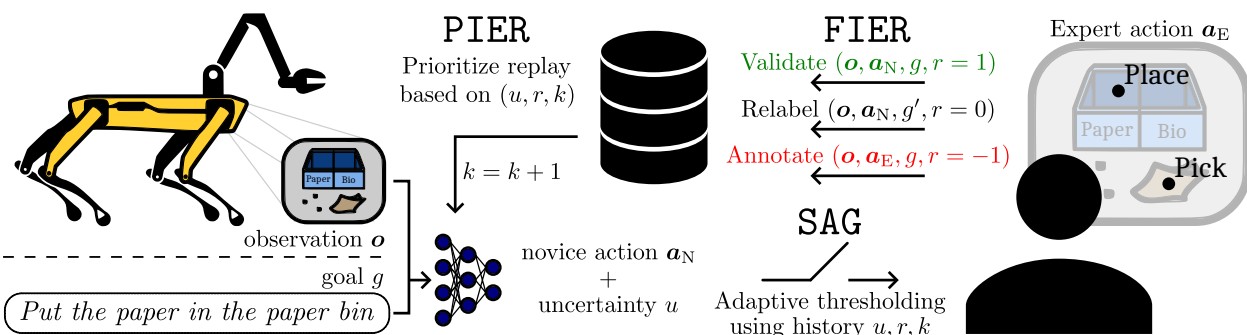

Figure 1: The Active Skill-level Data Aggregation (ASKDAGGER) framework consists of three main components: S-Aware Gating (SAG, detailed in Sec. 4.1), Foresight Interactive Experience Replay (FIER, detailed in Sec. 4.2), and Prioritized Interactive Experience Replay (PIER, detailed in Sec. 4.3). In this interactive imitation learning framework, we allow the novice to say: "*I plan to do this, but I am uncertain.*" The uncertainty gating threshold is set by SAG to track a user-specified metric: sensitivity, specificity, or minimum system success rate. This facilitates the trade-off between queries and failures. Teacher feedback is obtained with FIER, enabling demonstrations through validation, relabeling, or annotation demonstrations. Lastly, PIER prioritizes replay based on novice success, uncertainty, and demonstration age.

We make the following claims, considering an active data aggregation setting where the teacher can validate novice action plans:

**C1** SAG balances query count and system failures by tracking a user-specified metric value: desired sensitivity, specificity, or minimum system success rate.

**C2** FIER reduces the number of annotations needed to achieve a given success rate by recasting novice actions to demonstrations.

**C3** FIER enhances generalization to unseen scenarios by recasting failures to demonstrations.

**C4** PIER improves the success rate and reduces the required annotations under domain shift compared to uniform sampling.

The remainder of this paper is structured as follows. Sec. 2 reviews related work. The problem formulation is presented in Sec. 3. Our method is introduced in Sec. 4, followed by its experimental evaluation in Sec. 5. Sec. 6 discusses the results and limitations, and Sec. 7 concludes the paper.

## 2 Related Work

**Uncertainty-aware IIL:** In a seminal work on IIL with active learning, Chernova & Veloso (2007) introduced the Confidence-Based Autonomy (CBA) algorithm that combined the prediction confidence of a Gaussian Mixture Model (GMM) with the nearest neighbor distance from demonstration data to quantify the confidence of the novice policy. Based on this confidence measure, control is then gated between the novice policy and the human expert. Several related strategies exist, primarily as safety- and/or uncertainty-aware variants of the DAGGER algorithm (Ross et al., 2011), which we refer to as active DAGGER approaches. Like CBA, these methods apply a form of active learning, i.e., actively querying in situations deemed informative and/or risky. Such techniques include confidence measures based on prediction confidence (Grollman & Jenkins, 2007), maximum mean discrepancy (Kim & Pineau, 2013; Laskey et al., 2016), predicted proximity of novice actions to expert actions (Zhang & Cho, 2017), Monte Carlo dropout (Menda et al., 2017; Cui et al., 2019), ensembles (Menda et al., 2019; Li & Silver, 2023; Li & Zhang, 2023), variational autoencoder reconstruction error (Liu et al., 2024; Wong et al., 2021), value estimates (Hoque et al., 2022; Gokmen et al., 2023), ambiguity (Franzese et al., 2020; Luijkx et al., 2022), divergence (Datta et al., 2023), and diffusion policy training loss (Lee & Kuo, 2024). Outside the scope of IIL, robot-gating based on conformal prediction theory was introduced as well (Ren et al., 2023). In contrast to these robot-gated techniques, human-gated methods have also been proposed, requiring continuous human supervision, where the teacher

actively intervenes (Spencer et al., 2020; Kelly et al., 2019; Luo et al., 2024). There are also combinations of robot and human gating (Celemin & Kober, 2023; Hoque et al., 2022). Our approach differs from existing methods by considering the novice's actions during active queries, which we leverage in three ways. First, it enables a sensitivity-/specificity-/success-aware gating strategy to balance query frequency with error incidence while maintaining the desired metric value. Second, it allows novice actions to be recast as demonstrations by validating the novice's plan or relabeling the goal, inspired by Hindsight Experience Replay (HER) (Andrychowicz et al., 2017). Third, it enables replay prioritization based on novice success, drawing inspiration from Prioritized Experience Replay (PER) (Schaul et al., 2015). **Affordance Learning**: As ASKDAGGER learns goal-conditioned affordances in the form of skill parameters, it relates to affordance learning (Mo et al., 2021; Mazzaglia et al., 2024; Wang et al., 2022; Ning et al., 2023; Geng et al., 2023), which focuses on predicting where and how a robot can successfully apply a skill, aiming for generalization to unseen objects. Key differences are as follows. First, these affordance learning methods typically require reward signals for all interactions, whereas ASKDAGGER queries for demonstrations and requires teacher feedback only for robot-gated queries. Second, ASKDAGGER learns a policy providing goal-conditioned affordances, avoiding distribution mismatch by aligning training and policy observations. In contrast, existing affordance-learning methods primarily focus on efficiently exploring the large affordance state-action space rather than directly learning policies. Although these methods can learn from failures, they require extensive exploration and may experience distribution mismatch during deployment. **Reinforcement Learning from Human Feedback (RLHF):** As ASKDAGGER queries teacher demonstrations for uncertain novice actions, it is related to RLHF (Kaufmann et al., 2025). The most closely related approaches also use uncertainty-based querying, e.g., via information gain (Cui et al., 2019; Lindner et al., 2021). However, ASKDAGGER differs in that it requests new demonstrations when the teacher rejects a novice action, regulates the query rate through SAG, and collects relabeled demonstrations. In contrast, these RLHF methods typically learn a reward model and optimize the policy via reinforcement learning, whereas ASKDAGGER directly imitates expert actions. Another line of RLHF work focuses on human-gated methods, where teacher interventions are interpreted as negative rewards (Kahn et al., 2021; Luo et al., 2024). The main difference is that these methods require continuous human supervision during the data collection phase. Finally, Kumar et al. (2022) compare learning policies via behavioral cloning and offline RL from demonstrations. They conclude that the better choice depends on demonstration quality and the presence of critical states. Given our assumption of high-quality expert data and safety-critical decisions (e.g., skill misuse may damage the robot or its environment), our setting is better suited for a behavioral cloning approach. **Research gap:** Existing uncertainty-aware IIL approaches request expert queries based solely on uncertainty estimates. These gating methods do not take the novice's performance into account, as they discard the novice's actions when querying. In contrast, ASKDAGGER incorporates teacher feedback on queried actions and leverages this information for adaptive gating, demonstration collection, and prioritized replay for more efficient learning. Affordance learning methods aim to generalize to unseen objects but require constant reward information and are usually trained off-policy, which can lead to distribution mismatch at deployment. ASKDAGGER instead learns goal-conditioned affordances while collecting demonstrations on-policy, aligning training and execution distributions, while also not requiring constant reward information. RLHF methods often rely on learning a reward model or require continuous human supervision. ASKDAGGER avoids both by imitating expert actions and querying the teacher only when necessary, using validated or relabeled novice actions to enhance generalization and allowing for user-specified gating strategies.

## 3 Problem Statement

We consider an IIL problem, where a (robot) novice is learning interactively from (human) teacher feedback. The novice and teacher are denoted with subscripts N and T, respectively. The novice learns a policy $\pi_N : \mathcal{O} \times \mathcal{G} \to \mathcal{A}$ that maps observations $o_t^k \in \mathcal{O}$ and goals $g^k \in \mathcal{G}$ to actions $a_t^k \in \mathcal{A}$ during episode $k$ at time step $t$. We focus on mid- to high-level control tasks, where the novice has access to a set of predefined skills such as walking, grasping, or inserting. The novice learns from a demonstration dataset $\mathcal{D} = \{\tau^k\}_{k=0}^K$, consisting of trajectories $\tau$. These trajectories consist of the parameters of the demonstrated skills provided by the teacher. Contrasting with existing works, we let the teacher optionally provide a reward $r_t^k$ indicating whether the novice's actions were appropriate considering a goal $g^k$ and the observation $o_t^k$. Therefore, a

---

**Algorithm 1:** Active Skill-level DAgger (ASkDAgger)

---

**Input:** BC dataset $\mathcal{D}_{\mathrm{BC}}$, BC policy $\pi_{\mathrm{N}}^0$, teacher policy $\pi_{\mathrm{T}}$
**Parameters:** Gating mode $\mathtt{mode} \in \{\mathrm{sensitivity}, \mathrm{specificity}, \mathrm{success}\}$, desired gating mode value $\sigma_{\mathrm{des}}$, random query rate $p_{\mathrm{rand}}$, maximum number of episodes $k_{\mathrm{max}}$
**Output:** $\pi_{\mathrm{N}}^{k_{\mathrm{max}}}$

**1** $\boldsymbol{u} \leftarrow [\,], \boldsymbol{r} \leftarrow [\,], \boldsymbol{k} \leftarrow [\,], \mathcal{D} \leftarrow \mathcal{D}_{\mathrm{BC}}$
**2 for** *episode* $k = 0 : k_{\mathrm{max}} - 1$ **do**
**3**    $\boldsymbol{\tau}^k \leftarrow \emptyset, \boldsymbol{o}_0^k \leftarrow \mathtt{observe}(), g^k \leftarrow \mathtt{command}(), \mathtt{done} \leftarrow \mathtt{False}, t \leftarrow 0$
**4**    **while** not $\mathtt{done}$ **do**
**5**       $\boldsymbol{a}_t^k \leftarrow \pi_{\mathrm{N}}^k(\boldsymbol{o}_t^k, g^k)$
**6**       $u_t^k \leftarrow \mathtt{quantify\_uncertainty}(\pi_{\mathrm{N}}^k, \boldsymbol{o}_t^k, g^k)$
**7**       $\gamma \leftarrow \mathtt{SAG}(\boldsymbol{u}, \boldsymbol{r}, \boldsymbol{k}, \sigma_{\mathrm{des}}, p_{\mathrm{rand}}, \mathtt{mode})$                `// Set threshold to track `$\sigma_{\mathrm{des}}$` (Alg. 2)`
**8**       $\epsilon \sim U_{[0,1)}$
**9**       **if** $u_t^k \geq \gamma$ or $\epsilon < p_{\mathrm{rand}}$ **then**           `// Query actively and with probability `$p_{\mathrm{rand}}$
**10**          $\boldsymbol{a}_t^k, \boldsymbol{\tau}^k, r_t^k \leftarrow \mathtt{FIER}(\boldsymbol{o}_t^k, \boldsymbol{a}_t^k, \pi_{\mathrm{T}}, \boldsymbol{\tau}^k, g^k)$       `// Collect demonstration (Alg. 3)`
**11**          $\boldsymbol{o}_{t+1}^k, \mathtt{done} \leftarrow \mathtt{act}(\boldsymbol{a}_t^k)$
**12**       **else**
**13**          $\boldsymbol{o}_{t+1}^k, \mathtt{done} \leftarrow \mathtt{act}(\boldsymbol{a}_t^k)$
**14**       $t \leftarrow t + 1$
**15**    $\mathcal{D} \leftarrow \mathcal{D} \cup \boldsymbol{\tau}^k$
**16**    $P(i), \boldsymbol{w} \leftarrow \mathtt{PIER}(\boldsymbol{u}, \boldsymbol{r}, \boldsymbol{k})$                       `// Prioritize replay (Alg. 4)`
**17**    $\pi_{\mathrm{N}}^{k+1} \leftarrow \mathtt{update\_model}(\pi_{\mathrm{N}}^k, \mathcal{D}, P(i), \boldsymbol{w})$
**18 return** $\pi_{\mathrm{N}}^{k_{\mathrm{max}}}$

---

trajectory consists of tuples $\boldsymbol{\tau}^k = \{(\boldsymbol{o}_t^k, \boldsymbol{a}_t^k, g^k, r_t^k)\}_{t=0}^{T_k}$, where reward

$$r_t^k = \begin{cases} 1 & \text{if the teacher validates novice action;} \\ -1 & \text{if the teacher rejects novice action and provides an annotation;} \\ 0 & \text{otherwise.} \end{cases} \tag{1}$$

It is worth noting that $r_t^k$ is a teacher reward obtained during queries related to the *novice*'s actions, these actions may be different than $\boldsymbol{a}_t^k$. In our interactive approach, we collect data while executing the novice policy $\pi_{\mathrm{N}}^k$ and iteratively update it with the dataset $\mathcal{D}$, aggregating new demonstrations. Optionally, the policy can be pre-trained with a BC dataset $\mathcal{D}_{\mathrm{BC}}$. During this update, we aim to find the policy $\pi^*$ within policy space $\Pi$ that minimizes a loss measure $\mathcal{L}$ between the novice's actions and the teacher's actions, given the distribution of observations in $\mathcal{D}$: $\pi^* = \mathrm{argmin}_{\pi \in \Pi} \mathcal{L}(\pi, \mathcal{D})$. Since we generally do not have full state information, we consider $\boldsymbol{o}_t^k$ to result from an observation mapping $O : \mathcal{S} \to \mathcal{O}$ and we observe the state $\boldsymbol{o}_t^k = O(\boldsymbol{s}_t^k)$. We define the goal state set $\mathcal{S}^g \subset \mathcal{S}$ to be the set of states that satisfy the constraints of $g$. Therefore, an action $\boldsymbol{a}_t^k$ leads to $\mathtt{success}$ if $\boldsymbol{s}_{t+1}^k \in \mathcal{S}^g$. The set of states that result in achieving some goal is the union of all possible goal sets, i.e., $\mathcal{S}^G = \bigcup_{g \in \mathcal{G}} \mathcal{S}^g$. We assume the goal $g^k$ is constant throughout an episode, i.e., independent of the dynamics and actions taken. Therefore, a failure described by transition $(\boldsymbol{s}_t^k, \boldsymbol{a}_t^k, \boldsymbol{s}_{t+1}^k, g^k, \mathtt{success} = 0)$ can be "relabeled" to success $(\boldsymbol{s}_t^k, \boldsymbol{a}_t^k, \boldsymbol{s}_{t+1}^k, g', \mathtt{success} = 1)$ if $\boldsymbol{s}_{t+1}^k \in \mathcal{S}^G$, i.e., if the action resulted in achieving some other goal $g' \in \mathcal{G}$. So, if the teacher can observe $\boldsymbol{s}_t^k$ and (predict) $\boldsymbol{s}_{t+1}^k$, and can infer whether $\boldsymbol{s}_{t+1}^k \in \mathcal{S}^G$, the teacher can relabel failure transitions to successes. Furthermore, we consider an active approach based on the policy's prediction uncertainty. Therefore, we require an uncertainty operator $U : \Pi \times \mathcal{O} \times \mathcal{G} \to \mathbb{R}_{[0,1]}$ that provides the prediction uncertainty of the novice policy, given the current observation and goal, i.e., $u_t^k = U(\pi_{\mathrm{N}}, \boldsymbol{o}_t^k, g^k)$. Optionally, one can also take $\mathcal{D}$ into account when quantifying uncertainty, e.g., to quantify the proximity of an observation to those in $\mathcal{D}$.

## 4 Active skill-level Data Aggregation(ASkDAgger) Framework

ASkDAgger is an interactive imitation learning framework where the teacher is actively queried based on S-Aware Gating (SAG). The teacher can provide feedback through Foresight Interactive Experience Replay (FIER) in three modalities: validation, relabeling, or annotation demonstrations. We update the novice

policy using the demonstration dataset while we perform Prioritized Interactive Experience Replay (PIER). The main training procedure of ASKDAGGER, summarized in Alg. 1, follows these steps: In each episode $k$, at every time step $t$, the novice policy $\pi_N^k$ selects an action $\boldsymbol{a}_t^k$ based on observation $\boldsymbol{o}_t^k$ and goal $g^k$. Besides inferring its policy, the novice also quantifies the corresponding uncertainty $u_t^k$ (Alg. 1, lines 2-6). SAG then sets a gating threshold $\gamma$ to track the user-defined level of sensitivity, specificity or minimum system success rate $\sigma_{\text{des}}$ (line 7). The teacher is queried if the novice uncertainty exceeds the gating threshold. Additional queries are obtained with probability $p_{\text{rand}}$ to enhance the performance of the SAG algorithm (detailed in Sec. 4.1). During these queries the novice presents its planned action $\boldsymbol{a}_t^k$, allowing the teacher to validate, relabel, and/or provide an annotation demonstration (lines 9-11). If the teacher is not queried, the novice acts autonomously (lines 12-13). When the episode is `done`, e.g., because the goal constraints are satisfied or a time-out is reached, the demonstration trajectory $\boldsymbol{\tau}^k$ is added to $\mathcal{D}$. Finally, a model update can be performed using PIER (lines 16-17). Note that we keep track of the update/episode counts $\boldsymbol{k} = [0, \ldots, K_{T^K}]$, uncertainties $\boldsymbol{u} = [u_0^0, \ldots, u_T^K]$, and rewards $\boldsymbol{r} = [r_0^0, \ldots, r_T^K]$ during training with ASKDAGGER. The next sections will provide detailed descriptions of the subroutines from Alg. 1, i.e., SAG for gating (Alg. 2), FIER for demonstration collection (Alg. 3) and PIER for replay prioritization (Alg. 4).

## 4.1 S-Aware Gating (SAG)

At each time step, the uncertainty of the novice policy determines whether it should act autonomously or request teacher feedback. We introduce SAG for this gating problem. It dynamically adjusts the gating threshold $\gamma$ to maintain a user-specified target: sensitivity (`mode = sensitivity`), specificity (`mode = specificity`), or minimum system success rate (`mode = success`). In this context, queries are treated as positives and autonomous actions as negatives. A false positive occurs when the teacher is queried despite the novice's action being valid, which reduces autonomy. A false negative occurs when the teacher is not queried despite an invalid novice action, which results in system failure. Different gating modes are provided to suit varying task requirements. If system failures (false negatives) are costly, one can choose a high desired sensitivity. If unnecessary expert queries (false positives) are more costly, one can choose to ensure a high specificity. Finally, when the overall system success rate is the primary concern, the success mode allows specifying a minimum desired success rate. SAG continuously adjusts $\gamma$ to ensure the success rate meets this target. If the novice's success rate falls below the threshold, more queries are issued to increase reliability via expert interventions. If the success rate exceeds the threshold, queries are kept to a minimum, as the constraint is already satisfied.

We formalize the gating problem as a semi-supervised logistic regression, using uncertainty $u$ as the independent variable and reward $r$ as the indicator variable. The logistic regression assumption (the log-likelihood ratio of class distributions is linear in the observations) holds for various exponential distributions, such as normal, beta, and gamma distributions (Amini & Gallinari, 2002).

We summarize SAG in Alg. 2 and provide a visualization for more intuitive understanding in Fig. 2. For computing the gating threshold $\gamma$, we maintain a window of the most recent values in $\boldsymbol{u}_W, \boldsymbol{r}_W$, and $\boldsymbol{k}_W$ We do this because, with each model update, the uncertainty and reward information becomes more outdated (line 1 of Alg. 2). The window size is adjusted adaptively to ensure that the window contains at least $N_{\text{min}}$ relevant labels. Relevant labels correspond to $r = 1$ in specificity mode, $r = -1$ in sensitivity mode, and both are relevant in success mode. This is because, for instance, known failures (true positives) are essential when approximating sensitivity (true positive rate). Since the failure distribution over uncertainty shifts over time due to model updates, we normalize $\boldsymbol{u}_W$ (lines 2-4 of Alg. 2) to match the expected uncertainty at the current episode $K$. This is achieved by performing linear regression on $\boldsymbol{k}_W$ and $\boldsymbol{u}_W$, then adjusting for the expected difference in uncertainty between episode $k$ and $K$. This is visualized in Fig. 2 A. Next, we fit a logistic function to $-\boldsymbol{r}_W$ and $\boldsymbol{u}_W$ (line 5 of Alg. 2 and Fig. 2 B-C). We negate the rewards because, in the context of sensitivity, positive cases typically represent costly events (novice failures). We capture these failures by negating $\boldsymbol{r}_W$. If the teacher was not queried, we do not know whether the novice acted successfully at $(k, t)$. Since the uncertainty $u_t^k$ is known, we can obtain a pseudo-label by sampling from the fitted logistic model at $u_t^k$ (line 11 and Fig. 2 D). This enables us to approximate true and false positive rates for different threshold values.

---

**Algorithm 2:** S-Aware Gating (SAG)

---

**Input:** Uncertainties $\boldsymbol{u}$, rewards $\boldsymbol{r}$, update counts $\boldsymbol{k}$
**Parameters:** Gating mode $\texttt{mode} \in \{\text{sensitivity}, \text{specificity}, \text{success}\}$, desired value $\sigma_{\text{des}}$, random query rate $p_{\text{rand}}$,
          minimum number of negative labels $N_{\min}$, number of imputation repetitions $N_{\text{rep}}$
**Output:** Gating threshold $\gamma$

1   $\boldsymbol{u}_W, \boldsymbol{r}_W, \boldsymbol{k}_W \leftarrow \texttt{get\_window}(\boldsymbol{u}, \boldsymbol{r}, \boldsymbol{k}, N_{\min}, \texttt{mode})$
2   $w_{\text{lin}}, b_{\text{lin}} \leftarrow \texttt{LinRegres().fit}(\boldsymbol{k}_W, \boldsymbol{u}_W)$          `// Linear regression for normalizing u`
3   **for** $u_t^k \in \boldsymbol{u}_W$ **do**
4      $\lfloor \; u_t^k \leftarrow u_t^k + w_{\text{lin}}(K - k)$          `// Visualized in Fig. 2 A`
5   $w_{\text{log}}, b_{\text{log}} \leftarrow \texttt{LogRegres().fit}(\boldsymbol{u}_W, -\boldsymbol{r}_W)$     `// Logistic regression for imputation where` $r_t^k = 0$
6   $\boldsymbol{\gamma} \leftarrow [\,]$
7   **for** $i = 0 : N_{\text{rep}} - 1$ **do**
8      $\boldsymbol{f}_W \leftarrow -\boldsymbol{r}_W$          `// Visualized in Fig. 2 B`
9      **for** $r_t^k \in \boldsymbol{r}_W$ **do**
10        **if** $r_t^k == 0$ **then**          `// If teacher was not queried`
11          $\lfloor \; f_t^k \sim \{-1, 1\}, \quad P(f_t^k = 1) = \texttt{sigmoid}(w_{\text{log}} u_t^k + b_{\text{log}})$    `// Visualized in Fig. 2 C`
12      $\gamma_i \leftarrow \texttt{set\_threshold}(\boldsymbol{u}_W, \boldsymbol{f}_W, \sigma_{\text{des}}, p_{\text{rand}}, \texttt{mode})$          `// Visualized in Fig. 2 D`
13   $\gamma \leftarrow \texttt{median}(\boldsymbol{\gamma})$
14   **return** $\gamma$

---

Queries are not only made when uncertainty exceeds the threshold but also with probability $p_{\text{rand}}$. These random queries ensure that labels are collected across the entire uncertainty range, not just in the high-uncertainty regime. If random queries are not accounted for when setting the threshold, it will be too conservative, as more queries are triggered than intended. Since $p_{\text{rand}}$ is user-defined, we can incorporate these random queries when determining the gating threshold.

The overall SAG procedure is identical across modes, with the only difference being how the threshold is set (line 12). Here, we describe threshold selection for the sensitivity mode; for specificity- and success-aware gating, see App. A.1 and App. A.2, respectively. The total sensitivity $\sigma^{\text{sens}}$ can be computed by considering both the true positives $\text{TP}_\gamma$ and false negatives $\text{FN}_\gamma$ resulting from active gating, along with the true positives $\text{TP}_{\text{rand}}$ from random gating:

$$\sigma^{\text{sens}} = \frac{\text{TP}_\gamma + \text{TP}_{\text{rand}}}{\text{TP}_\gamma + \text{FN}_\gamma}. \tag{2}$$

The number of $\text{TP}_{\text{rand}}$ are determined by which of $\text{FN}_\gamma$ are queried. These queries occur with probability $p_{\text{rand}}$ (see lines 8-9 of Alg. 1). Therefore, its expected value is $\text{FN}_\gamma \cdot p_{\text{rand}}$. This leads to:

$$\mathbb{E}_{\epsilon \sim U_{[0,1)}}[\sigma^{\text{sens}}] = \frac{\text{TP}_\gamma}{\text{TP}_\gamma + \text{FN}_\gamma} + \frac{\text{FN}_\gamma \cdot p_{\text{rand}}}{\text{TP}_\gamma + \text{FN}_\gamma} \tag{3}$$

$$= \sigma_\gamma^{\text{sens}} + p_{\text{rand}}(1 - \sigma_\gamma^{\text{sens}}), \tag{4}$$

where $\sigma_\gamma^{\text{sens}}$ is the sensitivity for gating disregarding the random queries. Thus, we interpolate between threshold values that best satisfy the desired sensitivity level $\sigma_{\text{des}} = \sigma_\gamma^{\text{sens}} + p_{\text{rand}}(1 - \sigma_\gamma^{\text{sens}})$ (line 12 of Alg. 2 and Fig. 2 D).

For each mode, the process of sampling pseudo labels is repeated $N_{\text{rep}}$ times, and the final threshold $\gamma$ is set as the median of the thresholds obtained across repetitions.

## 4.2 Foresight Interactive Experience Replay (FIER)

When the novice's uncertainty exceeds the threshold set by SAG, feedback is requested from the teacher through FIER. This method enhances demonstration collection by considering the novice's planned actions during queries and presenting them to the human teacher. The FIER procedure is summarized in Alg. 3. FIER reduces the number of required teacher annotations by enabling the human teacher to provide two additional feedback modalities. First, we allow the teacher to validate the proposed actions. If the teacher

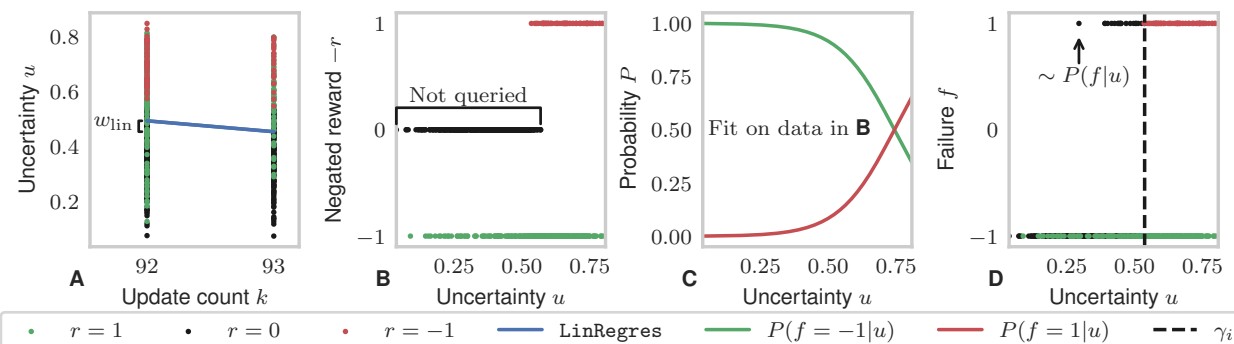

Figure 2: Visualization of the SAG algorithm (mode = sensitivity) with experimental data from Sec. 5.1. First, we normalize the uncertainty values in $\boldsymbol{u}_W$ using linear regression, as the uncertainty distribution shifts with the number of updates **(A)**. By negating the rewards $\boldsymbol{r}_W$, we obtain labels for novice failures, denoted as $\boldsymbol{f}_W$ **(B)**. Labels are unavailable when the teacher was not queried ($r = 0$). However, since uncertainties at those time steps are known, we generate pseudo labels by sampling from a logistic distribution fit to $\boldsymbol{u}_W$ and $\boldsymbol{f}_W$ **(C)**. We then compute a gating threshold $\gamma_i$ using both the labels and pseudo labels **(D)**. This sampling and threshold calculation process is repeated $N_{\text{rep}}$ times.

considers the plan valid ($r == 1$ in line 2 in Alg. 3), we execute and add the novice's planned actions to the demonstration dataset. The novice plan can be valuable even if it is invalid. Inspired by HER Andrychowicz et al. (2017), we can utilize invalid novice plans as demonstrations if they achieve another goal (line 7 in Alg. 3). In this case, we can obtain a new demonstration by letting the teacher relabel the goal. The benefit of relabeling demonstrations is twofold. First, there can be situations where relabeling the goal is less demanding for the teacher than providing an annotation. In that case, it is possible to collect additional demonstrations at a reduced cost. Secondly, it allows for the collection of demonstrations for goals induced by the novice policy instead of collecting demonstrations only for goals induced by the distribution of commands. This way, the novice can learn to perform tasks beyond the instructed commands and possibly generalize better to novel scenarios. After providing the option to relabel the novice's actions, the teacher is asked to provide an annotation demonstration (line 6). To summarize, we can collect three types of demonstrations with FIER: validation, relabeling, and annotation demonstrations.

### 4.3   Prioritized Interactive Experience Replay (PIER)

At the end of the episode, the demonstrations collected through FIER are aggregated into the demonstration dataset, allowing the novice policy to be updated. Efficiently performing policy updates is particularly important in interactive imitation learning, as this paradigm involves a human teacher providing online feedback. Taking inspiration from PER (Schaul et al., 2015), we introduce an interactive equivalent, which we call PIER (summarized in Alg. 4). PIER prioritizes the replay of the demonstration dataset based on uncertainty, novice success, and demonstration age. We prioritize demonstrations where the novice fails over successes. Those with low uncertainty receive the highest priority among failures, as they suggest confident yet mistaken actions. While among successes, those with high uncertainty are prioritized to reduce the novice's uncertainty for those situations. Successes with low uncertainty, indicating proficient performance, are given the lowest priority. Since the uncertainty and novice success information become outdated with each model update, we diminish the prioritization based on the age of the demonstration. Alg. 4 shows how these desired properties are integrated into our prioritized replay scheme. Similar to PER (Schaul et al., 2015), we define the probability of sampling demonstration tuple of episode $k$ at timestep $t$ to be:

$$P(k, t) = \frac{(p_t^k)^\alpha}{\sum_i \sum_j (p_j^i)^\alpha}. \tag{5}$$

Here $p_t^k$ is the priority of the demonstration tuple from episode $k$ at time step $t$ and $\alpha \geq 0$. Increasing $\alpha$ results in more prioritization, while $\alpha = 0$ corresponds to uniform sampling. We define the prioritization exponent

| **Algorithm 3:** Foresight Interactive Experience Replay (FIER) | **Algorithm 4:** Prioritized Interactive Experience Replay (PIER) |
|---|---|
| **Input:** Observation $o$, novice action $a$, teacher policy $\pi_\mathrm{T}$, trajectory $\tau$, goal $g^k$ 
 **Parameters:** Goal set $\mathcal{G}$ 
 **Output:** Action $a$, trajectory $\tau$ | **Input:** Uncertainties $u$, rewards $r$, update counts $k$ 
 **Parameters:** Scale $\lambda$, base $b$, exponents $\alpha, \beta$ 
 **Output:** Sampling priorities $p$, weights $w$ |
| **1** $r, g' \leftarrow \mathrm{query}(o, a)$ 
 **2 if** $r == 1$ **then**          // Validation tuple 
 **3** $\quad \lfloor\ \tau \leftarrow \tau \cup (o, a, g = g^k, r = 1)$ 
 **4 else** 
 **5** $\quad a \leftarrow \pi_\mathrm{T}(o)$          // Annotation tuple 
 **6** $\quad \tau \leftarrow \tau \cup (o, a, g = g^k, r = -1)$ 
 **7** $\quad$ **if** $g' \in \mathcal{G}$ **then**          // Relabeled tuple 
 **8** $\quad\quad \lfloor\ \tau \leftarrow \tau \cup (o, a, g = g', r = 0)$ 
 **9 return** $a, \tau, r$ | **1** $w = [\ ]$ 
 **2 for** $k = 0 : K$ **do** 
 **3** $\quad$ **for** $t = 0 : T^k$ **do** 
 **4** $\quad\quad c_t^k = \lambda u_t^k + (1 - \lambda) \frac{K - k}{K}$ 
 **5** $\quad\quad p_t^k \leftarrow 1 - r_t^k \frac{b^{1 - c_t^k} - 1}{b - 1}$ 
 **6** $\quad\quad P(k, t) \leftarrow \frac{(p_t^k)^\alpha}{\sum_i \sum_j (p_j^i)^\alpha}$ 
 **7** $\quad\quad w_t^k = (|k| \cdot P(k, t))^{-\beta} / \max_{i,j} w_j^i$ 
 **8 return** $P(k, t), w$ |

as a linear combination of the uncertainty and the number of model updates since the demonstration was added to the dataset, i.e., $c_t^k = \lambda u_t^k + (1 - \lambda)(\frac{K-k}{K})$. Here, $0 \le \lambda \le 1$ scales prioritization based on the uncertainty versus novelty of the sample. Finally, the priorities are:

$$p_t^k = 1 - r_t^k \frac{b^{1-c_t^k} - 1}{b - 1}, \tag{6}$$

where $b > 1$ is the base. In this way, the priorities of old demonstrations with high novice uncertainty depend little on novice success. In contrast, the priorities for recent demonstrations with low novice uncertainty depend greatly on novice success. Having defined the priorities, we need to compensate for the bias that prioritization introduces when minimizing the expectation of a loss function $l$, i.e., $\mathbb{E}_{k,t \sim P(k,t)} \mathcal{L}\left(\pi_\mathrm{N}, (o_t^k, a_t^k)\right)$, instead of sampling $k, t$ according to the distribution of the dataset—i.e., $k, t \sim D(k, t)$. Again, following PER (Schaul et al., 2015), we can mitigate the bias by introducing importance-sampling weights $w = [w_0^0, \dots]$ (see line 7 of Alg. 4). Here, $\beta$ determines the level of bias compensation. The weights are normalized to prevent numerical instabilities.

## 5 Experimental Evaluation

To support claims **C1-4** from Sec. 1, we evaluate ASKDAGGER and its components in four sets of experiments. First, we performed active dataset aggregation on the MNIST dataset (LeCun et al., 1998) using TorchUncertainty (Lafage & Laurent, 2024) to validate SAG extensively. Second, we interactively trained CLIPort agents on simulated language-conditioned tabletop manipulation tasks (Shridhar et al., 2021). Third, we conducted experiments on a real-world assembly setup to demonstrate that these claims extend beyond simulation. Finally, we showcase ASKDAGGER's applicability by integrating it with built-in primitive actions on a Spot robot to perform a sorting task.

### 5.1 MNIST Dataset Aggregation

To support claim **C1** — *SAG balances query count and system failures by tracking a user-specified metric value: desired sensitivity, specificity, or minimum system success rate* — we conducted experiments in which we interactively trained digit classification models on the MNIST dataset (LeCun et al., 1998) [1]. We selected this setup due to its low computational requirements, enabling extensive ablations and easy reproducibility. Additionally, existing packages such as TorchUncertainty (Lafage & Laurent, 2024) facilitate uncertainty quantification in this setting. Since we focus on the SAG component, we follow the procedure described in Alg. 1, but without demonstration collection via relabeling or replay prioritization. To validate whether SAG can track a desired sensitivity, we performed interactive training for nine different sensitivity, specificity and success rate values, i.e., $\sigma_\mathrm{des} \in \{0.1i\}_{i=1}^9$, repeating the procedure ten times for each value of $\sigma_\mathrm{des}$.

---

[1]The code and data from these experiments are available at `https://github.com/askdagger/askdagger_mnist`.

These experiments proceed as follows. We sample a batch of 128 handwritten digit images from the MNIST dataset at each timestep without replacement. This allows for 468 timesteps, as the dataset contains 60,000 samples. Although MNIST provides ground truth labels, we simulate an active learning scenario where labels are queried if the prediction uncertainty exceeds the threshold $\gamma$ set by SAG or randomly with probability $p_{\text{rand}} = 0.1$. Uncertainty quantification is performed using Monte Carlo Dropout (MC Dropout) (Gal & Ghahramani, 2016) with a dropout rate of 0.4 and 16 stochastic forward passes, forming an ensemble $\mathcal{C} = \{h_1, \ldots, h_{16}\}$. For a sample $x$ with label $y$, prediction uncertainty is computed as $u = 1 - \max_y P_{\mathcal{C}}(y|x)$, where $P_{\mathcal{C}}(y|x) = \frac{1}{16}\sum_{i=1}^{16} P_i(y|x)$. Ground truth labels are obtained for the queried samples and added to the training dataset. The model is updated every five timesteps using the aggregated dataset.

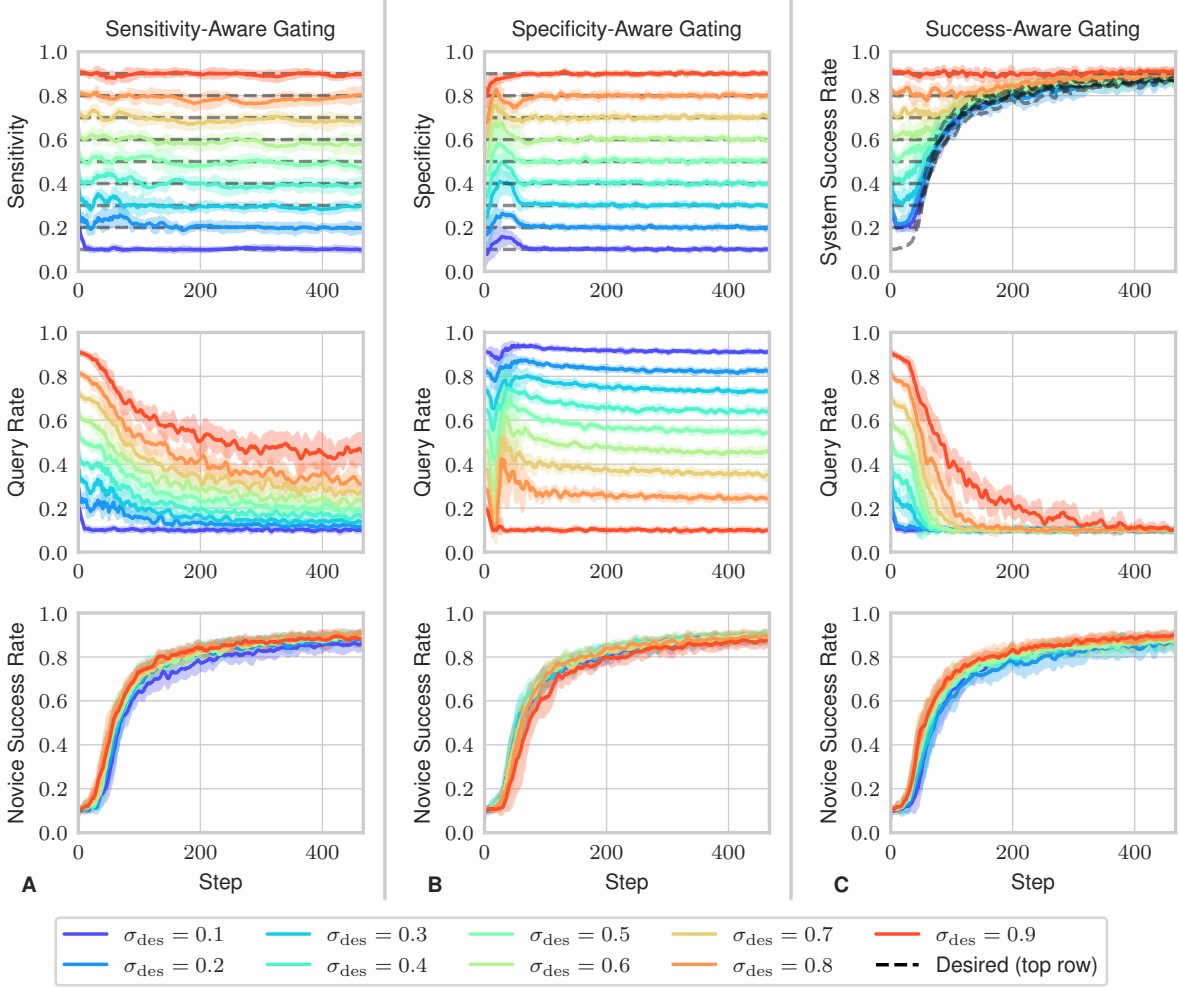

Figure 3: Results for various levels of $\sigma_{\text{des}}$ for sensitivity-aware (**A**), specificity-aware (**B**), and success-aware (**C**) gating in active dataset aggregation with SAG on the MNIST (LeCun et al., 1998) dataset. Sensitivity in **A** and specificity in **B** are calculated over a moving window of 1000 failures and successes, respectively. Novice and system success rates are calculated over a window of 1000 samples (approximately eight steps). Mean and standard deviation are shown for ten repetitions.

The results of these experiments are summarized in Fig. 3. The sensitivity and specificity plots in Fig. 3 A and B show that SAG successfully tracks the desired levels for all nine values of $\sigma_{\text{des}}$. In success-aware mode, Fig. 3 C shows that when the novice success rate is low, SAG issues enough queries to maintain the desired system success rate $\sigma_{\text{des}}$. As the novice success rate increases, the query rate decreases, reaching a minimum once the novice success rate exceeds $\sigma_{\text{des}}$. The query rate plots in Fig. 3 also indicate that each

mode requires a different query pattern to track its respective metric. The success rate plots show that, in all modes, the novice ultimately learns to perform the task.

## 5.2 CLIPort Benchmark Tasks

To support claims **C1-3**, we conducted experiments using ASĸDAGGER to train CLIPort (Shridhar et al., 2021) agents interactively [2]. CLIPort is a language-conditioned imitation-learning agent for vision-based manipulation. The observations in these experiments consist of an RGB-D image and a natural language text command. CLIPort employs a two-stream architecture: a spatial stream and a semantic stream. The semantic stream uses frozen CLIP encoders (Radford et al., 2021) to extract features from the RGB image and the language command. The spatial stream is an untrained Transporter network (Zeng et al., 2021), whose decoder layers are fused with features from the semantic stream. The model outputs pixel-wise value estimates for both picking and placing. The demonstration dataset includes RGB-D images, language commands, and expert actions in the form of Cartesian pick and place poses, represented as a pixel location with discretized orientation. We selected this setup because it allows novices to communicate their actions by indicating planned pick-and-place locations on an image alongside a language command, making it well-suited for ASĸDAGGER.

We compared ASĸDAGGER's performance against an active DAGGER baseline without both PIER and FIER to provide evidence for claims **C2-3**. We also compare ASĸDAGGER against SafeDAGGER (Zhang & Cho, 2017) and ThriftyDAGGER (Hoque et al., 2022), which are also DAGGER approaches that incorporate active learning. Furthermore, we performed ablations with ASĸDAGGER without PIER and ASĸDAGGER without FIER to isolate the effects of the individual components. The models trained with ASĸDAGGER and active DAGGER use SAG for gating and rely on prediction entropy to quantify uncertainty for a fair comparison.

The comparison was conducted across a subset of tasks by Shridhar et al. (2021). These tasks are visualized in Fig. 4. The CLIPort benchmark includes *seen* and *unseen* task settings. In the *unseen* setting, test-time commands involve different objects, shapes, or colors than during training. For example, $\mathbb{T}_{\text{seen colors}} = \{\text{yellow}, \text{brown}, \text{gray}, \text{cyan}\}$ and $\mathbb{T}_{\text{unseen colors}} = \{\text{purple}, \text{pink}, \text{white}, \text{black}\}$. Some colors appear in both settings, i.e., $\mathbb{T}_{\text{all colors}} = \{\text{red}, \text{green}, \text{blue}\}$. A complete list of *seen* and *unseen* objects is provided in Tab. 4 in App. A.8. We modified the tasks involving Google objects and shapes by sometimes introducing *unseen* objects as distractors during training, as real-world scenarios also involve varying distractor objects. Therefore, it is possible for ASĸDAGGER to acquire demonstrations for the *unseen* set via relabeling with FIER when the novice fails in a meaningful way. This setup allows us to provide evidence for claim **C3** by evaluating whether failures can be relabeled and whether this improves generalization to the *unseen* setting.

The following hyperparameters were used for both ASĸDAGGER and the active DAGGER baseline if applicable. For SAG we used $\text{mode} = \text{sensitivity}$, $\sigma_{\text{des}} = 0.9, N_{\min} = 15, p_{\text{rand}} = 0.2$ and for PIER $\alpha = 1.5, b = 10, \beta = 1$ and $\lambda = 0.5$. Each setting involved training ten CLIPort agents without BC pretraining, collecting 300 interactive demonstrations, and evaluating checkpoints every 100 demonstrations. Model updates occur at the end of an episode if a demonstration is collected. This way, we ensure that ASĸDAGGER and the baselines had the same number of updates.

The cumulative rewards for evaluating checkpoints on tasks with *seen* and *unseen* objects are shown in Fig. 5. While ASĸDAGGER performs equally or better on all tasks, the number of teacher annotations is significantly lower, as shown in Fig. 6. The performance gain of ASĸDAGGER can be attributed to the composition of the demonstration dataset. For the active DAGGER baselines, all demonstrations consist of annotation tuples, whereas ASĸDAGGER collects many through validation and relabeling. The relabeled demonstrations explain ASĸDAGGER's superior performance on *unseen* tasks: agents sometimes obtained demonstrations by relabeling novice failures, where the intended pick was a distractor object from the *unseen* set. Fig. 6 also shows that ASĸDAGGER requires fewer annotation demonstrations than the active DAGGER baselines. The corresponding sensitivity curves for ASĸDAGGER in Fig. 7 confirm that SAG maintains the desired sensitivity level across all tasks.

---

[2]The code, data, videos, and a notebook are available at `https://github.com/askdagger/askdagger_cliport`.

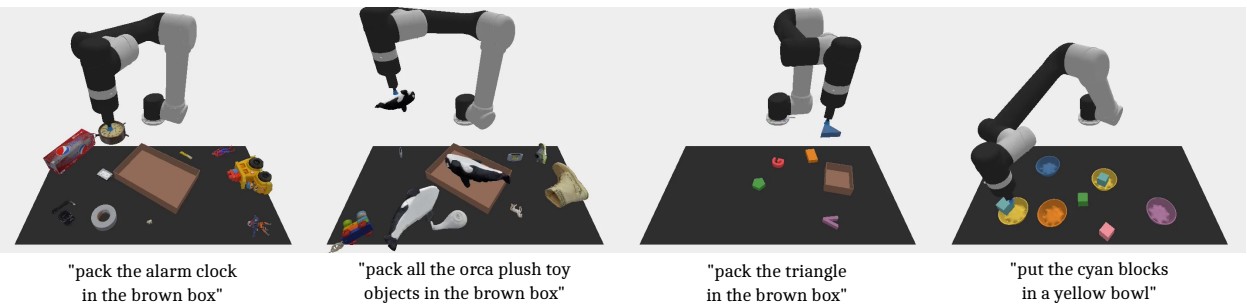

Figure 4: The CLIPort benchmark tasks from left to right: *packing-google-objects-seq*, *packing-google-objects-group*, *packing-shapes* and *put-blocks-in-bowls*.

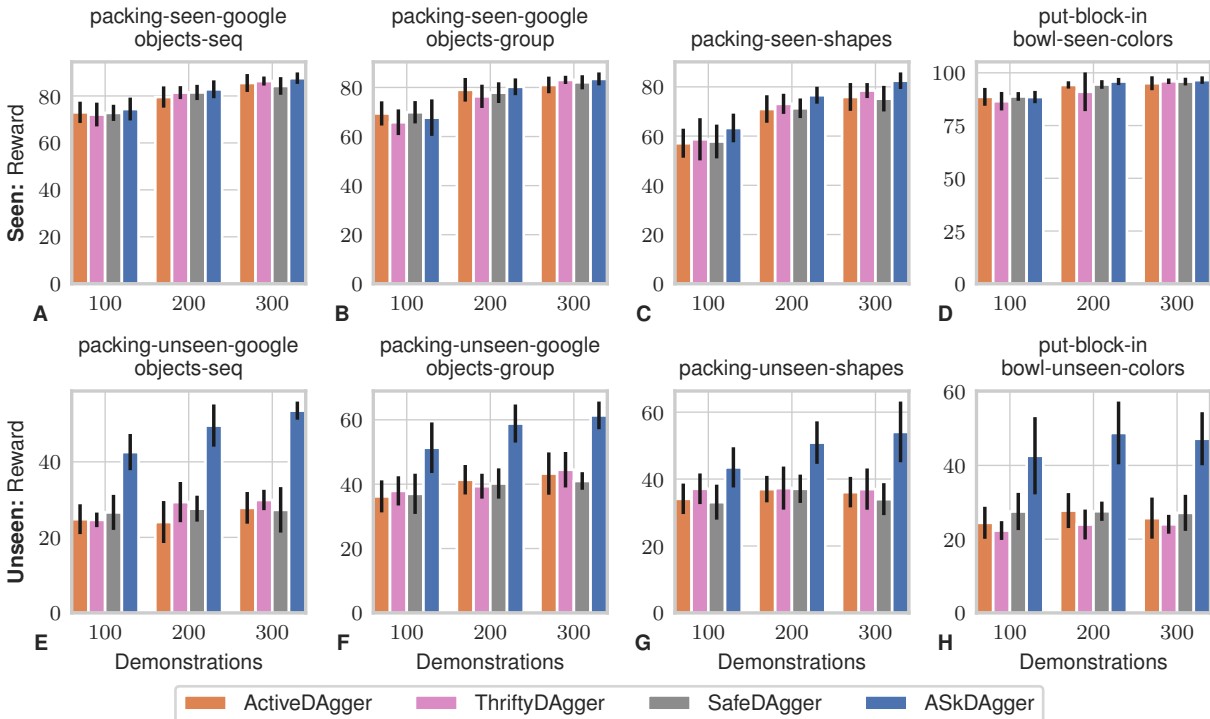

Figure 5: Cumulative rewards for evaluating checkpoints over 100 episodes on tasks with *seen* and *unseen* objects. Mean and standard deviation are shown for ten policies using a moving window of 50 episodes. The results show clear improvements with ASKDAGGER on the *unseen* scenarios.

To support **C4**, we performed ablations with ASKDAGGER under domain shifts. We trained agents using ASKDAGGER, ASKDAGGER without PIER (w/o PIER), and ASKDAGGER without FIER (w/o FIER) on a sequence of tasks with increasing domain shifts: *packing-seen-shapes*, *packing-unseen-shapes*, and *packing-seen-google-objects-seq*. As shown in Fig. 8, ASKDAGGER and ASKDAGGER w/o PIER perform similarly on the initial task, while ASKDAGGER w/o FIER performs slightly worse, likely due to the benefits of relabeling demonstrations. Moreover, ASKDAGGER w/o FIER requires more annotation demonstrations (Fig. 8 D). After transitioning to *packing-unseen-shapes*, ASKDAGGER adapts slightly faster to the domain shift initially, though performance is similar after 300 demonstrations across all settings. This improved adaptation likely stems from FIER's relabeling and PIER's replay prioritization. Upon shifting to *packing-seen-google-objects-seq*, ASKDAGGER and ASKDAGGER w/o FIER outperform ASKDAGGER w/o PIER, highlighting the benefits of replay prioritization with PIER. This effect is more pronounced after the second transition, as the shift from *unseen-shapes* to *seen-google-objects-seq* is larger than from *seen* to *unseen* shapes. Additionally, as

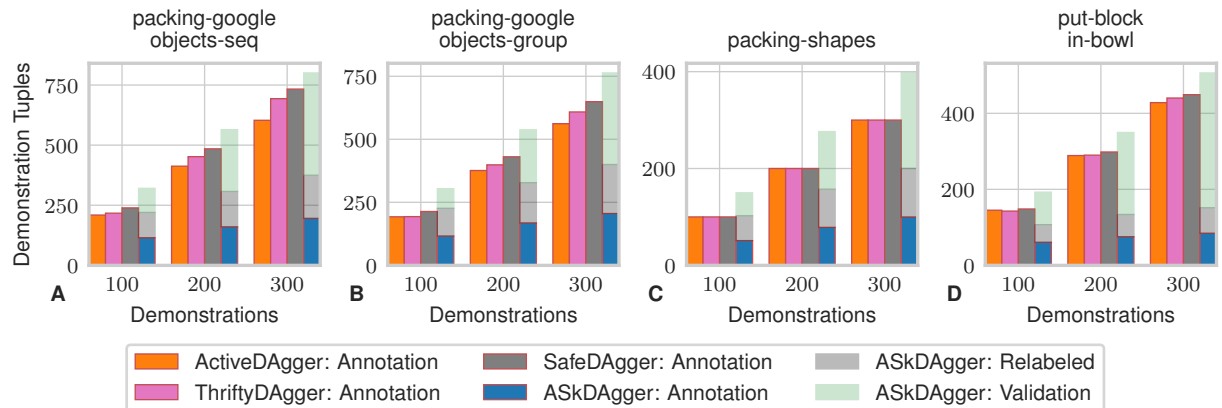

Figure 6: Composition of the demonstration datasets, showing mean values for ten policies. ASkDAGGER relies on fewer annotation demonstrations and benefits from relabeling and validation demonstrations.

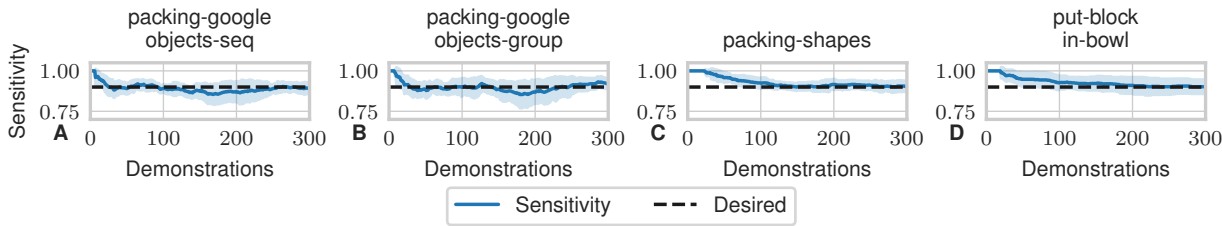

Figure 7: Sensitivity during training for ASkDAGGER on the CLIPort tasks. Sensitivity is calculated over a moving window of 50 failures. Mean and standard deviation are shown for ten policies.

the demonstration dataset grows, the probability of sampling a specific demonstration decreases for uniform sampling, further increasing PIER's effect. Finally, ASkDAGGER's improved performance on the third task requires fewer annotation demonstrations, as more validation demonstrations are collected (Fig. 8 F).

### 5.3 Real-World Engine Assembly

We conducted experiments on a real-world assembly task to demonstrate that our claims extend beyond simulation and showcase ASkDAGGER's applicability in real-world settings [3]. This task is a simplified version of a diesel engine assembly using 3D-printed models. The procedure is illustrated in Fig. 9. As shown in Fig. 9 A, the setup includes a Franka Panda robot equipped with an in-hand RealSense D405 RGB-D camera and a Franka hand with custom-printed fingers for grasping bolts. The control scheme is implemented using the EAGERx framework (van der Heijden et al., 2024). The objective is to pick bolts from a holder and insert them into specific locations on the engine block. We use pick-and-place primitives that rely on 2D Cartesian positions, assuming a given height for picking and placing.

The task involves four bolt colors (red, yellow, green, and black) and seven insertion locations. The bolts are randomly ordered and placed in a holder. The human operator interacts with the robot via the interface shown in Fig. 9 B. This Gradio (Abid et al., 2019) interface allows command input via speech or text. In our experiments, we generate random commands in the form: "*Insert the* [color] *bolt at location number* [location number]*.*" Upon receiving a command, a top-down RGB-D reconstruction is obtained following Zeng et al. (2021), and the novice policy is evaluated. We use a CLIPort agent with the same hyperparameters as in Sec. 5.2, but with a smaller batch size of three due to GPU memory constraints on our laptop. If the prediction uncertainty exceeds the threshold $\gamma$ set by SAG, or randomly with probability $p_{\text{rand}} = 0.2$, the robot queries the human operator. This is shown in Fig. 9 C. The robot's planned action is visualized in the

---

[3]A video of this experimental evaluation is available at `https://askdagger.github.io`.

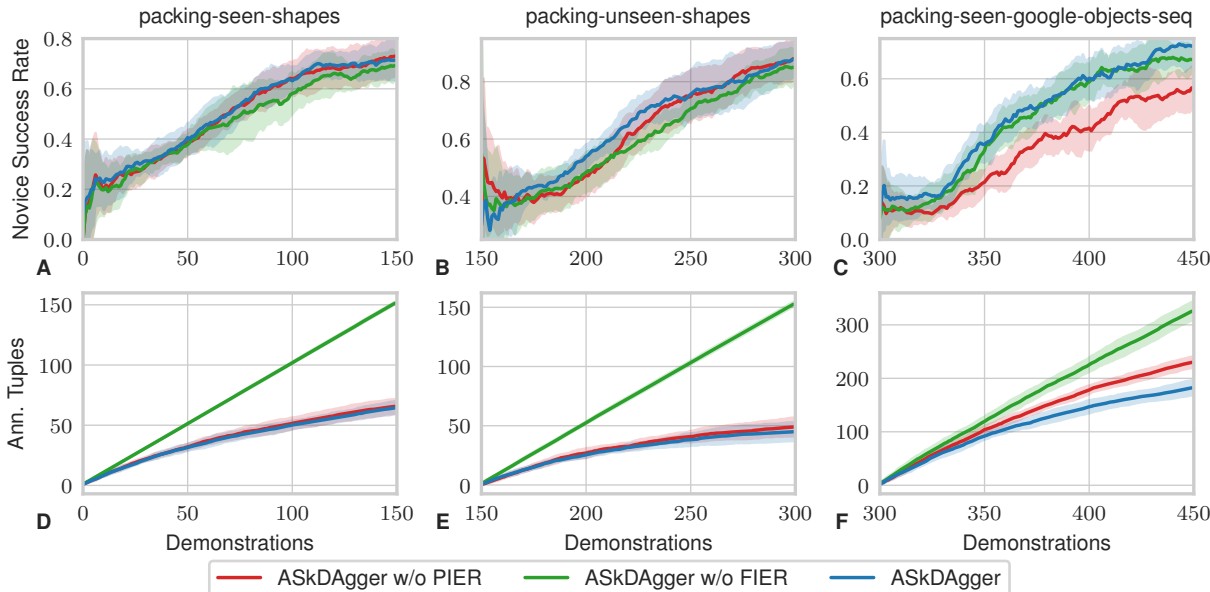

Figure 8: Novice success rate during training (**A-C**) and the number of collected annotation tuples (**D-F**) under domain shifts. The success rate is computed over a moving window of 50 episodes, reinitialized after each domain shift. The first 150 demonstrations are collected on *packing-seen-shapes*, the next 150 on *packing-unseen-shapes*, and the final 150 on *packing-seen-google-objects-seq*. Mean and standard deviation are shown for ten policies.

interface, allowing the operator to validate, relabel, or reject the plan. If validated, the robot executes the plan and it is added to the dataset. If relabeling is required, as in Fig. 9 C, the operator is prompted to provide corrections by selecting the correct bolt and location. This is shown in Fig. 9 D. The robot then executes the annotation demonstration (Fig. 9 F) and aggregates the relabeling and annotation demonstrations into the dataset. When a demonstration is collected, a model update is performed while the robot executes the demonstration.

The results for collecting 150 demonstrations are summarized in Fig. 10. The novice and system success rates in Fig. 10 A show that while the novice is learning to perform the task, the system can maintain a high success rate by querying based on SAG. The composition of the dataset in Fig. 10 B shows a similar trend as in the simulated experiments. This figure shows that ASkDAgger can learn from mostly validation demonstrations in the later stages of training. Also, it shows that relabeling demonstrations is possible not only in simulation but also in real-world scenarios. The sensitivity in Fig. 10 C shows that SAG can track the desired sensitivity level across the task.

### 5.4 Real-World Sorting Task

To further demonstrate ASkDAgger's applicability in real-world scenarios, we integrated it with Boston Dynamics Spot quadruped's built-in primitive skills to perform a sorting task [4]. Since ASkDAgger is designed to work with any robot with one or more skills, we selected Spot for its built-in grasping, walking, and placing capabilities. The task involves sorting objects into paper and organic waste bins, as illustrated in Fig. 11. For this demonstration, we trained a CLIPort agent using an interface similar to Fig. 9 with the command format: "*Put the* [object] *in the* [bin type] *bin.*" Since CLIPort requires a top-down RGB-D projection, we use the in-hand RGB-D camera to scan the environment from different perspectives. With these images, we can obtain a top-down projection using the robot's joint states, odometry, and camera intrinsics. The teacher issues the command through a Gradio (Abid et al., 2019) interface at the beginning of an episode. Similar to Sec. 5.3, the teacher is queried if the novice's uncertainty exceeds the gating

---

[4]A video of this demonstration is available at `https://askdagger.github.io`.

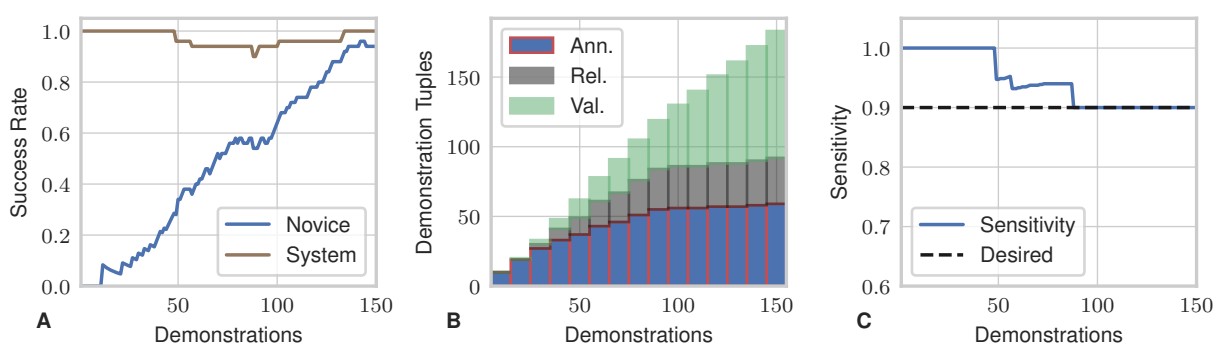

Figure 9: Real-world implementation of ASĸDAGGER on an engine assembly task. **(A)** The setup includes a Franka Panda robot, an RGB-D camera, and 3D-printed parts. **(B)** The interface allows the operator to issue commands. **(C)** When queried, the operator can validate, relabel, or reject the plan. **(D)** The operator relabels the plan. **(E)** The operator provides an annotation demonstration. **(F)** The robot executes the demonstration.

Figure 10: Results for the engine assembly task. **(A)** Success rates calculated over a window of 50 episodes. **(B)** Composition of the demonstration dataset. **(C)** Sensitivity calculated over a moving window of 50 failures.

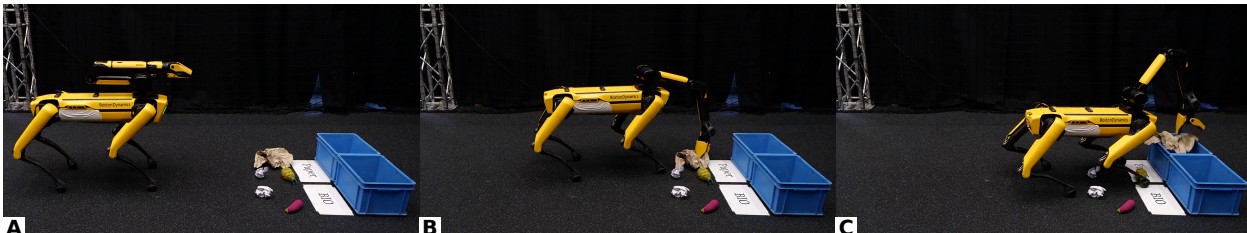

Figure 11: Real-world implementation of ASKDAGGER on a sorting task with Spot **(A)**. This demonstrates ASKDAG-GER's applicability in scenarios where a robot has access to one or more skills, such as grasping **(B)**, walking, and placing **(C)**.

threshold, and the interface shows the planned pick and place locations. The teacher can choose to validate or relabel the novice's actions. The teacher can provide annotation demonstrations by clicking where to pick and where to place in the top-down projected image. Demonstrations are aggregated to the demonstration dataset at the end of the episode, and the policy is updated. In the meantime, the robot walks back to its initial position and rescans its environment.

## 6 Discussion

This section discusses the results from Sec. 5 by revisiting the claims in Sec. 1. We also examine the limitations of the ASKDAGGER framework and its experimental evaluation.

### 6.1 Claims

Our experiments were designed to provide evidence for claims **C1-4** from Sec. 1. First, we claim that *SAG balances query count and system failures by tracking a user-specified metric value: desired sensitivity, specificity, or minimum system success rate* (**C1**). We provide evidence for this claim by showing accurate sensitivity, specificity, and system success rate tracking for multiple values of $\sigma_{\mathrm{des}}$ on an MNIST dataset aggregation task in Fig. 3. Moreover, we extend these results for the sensitivity-aware mode to simulated and real-world robot tasks in Sec. 7 and Fig. 10 C. Secondly, we claim that *FIER reduces the number of annotations needed to achieve a given success rate* (**C2**). We trained CLIPort agents on a set of benchmark tasks in simulation for 300 demonstrations and evaluated the performance of checkpoints saved every 100 demonstrations. The results from these experiments support this claim. The evaluation results in Fig. 5 show that ASKDAGGER performs equivalent or better compared to SafeDAGGER (Zhang & Cho, 2017), ThriftyDAGGER (Hoque et al., 2022), and ActiveDAGGER, while Fig. 6 shows that ASKDAGGER requires significantly fewer teacher annotations. The results in Fig. 10 provide evidence that these results generalize to real-world tasks, as we see a similar composition of the dataset and similar performance improvements. Thirdly, we claim that *FIER enhances generalization to unseen scenarios by recasting failures to demonstrations* (**C3**). The results in Fig. 5 provide evidence for this claim, as the ASKDAGGER checkpoints outperform the active baselines on the unseen scenarios. This can be attributed to relabeling failures involving the distractor objects, resulting in demonstrations for the unseen scenarios. Finally, we claim that PIER improves the success rate and reduces the required annotations under domain shift compared to uniform sampling. Evidence for this claim is provided by the results in Fig. 8, where PIER outperforms uniform sampling under domain shift regarding success rate while requiring fewer annotations.

### 6.2 Limitations

ASKDAGGER is designed for tasks with sparse rewards and learning mid- to high-level control. While this covers many problems, it does not extend to applications requiring high-rate feedback from the teacher. Thus, ASKDAGGER is best suited for scenarios where a robot has access to predefined skills and can learn higher-level plans for these skills. Additionally, ASKDAGGER can be integrated with methods that are better suited for learning low-level control or longer-horizon reasoning. While FIER significantly improves

performance, it relies on recasting failures as successes, which can be challenging in some applications. Additionally, although ASKDAGGER reduces the number of teacher annotations, it assumes the teacher can validate or relabel actions before execution. This may not always be feasible. In such cases, relabeling and validation can still occur post-execution. The impact of ASKDAGGER on teaching effort depends on both the task and the teaching interface. Nonetheless, we provide an in-depth discussion in Sec. A.6, including timing measurements from Sec. 5.3 and cost scenarios for Sec. 5.2. Tuning hyperparameters in real-world settings is challenging. However, our experiments show that a single set of hyperparameters, without extensive tuning, generalizes effectively across different tasks, suggesting that hyperparameter sensitivity is not strongly dependent on task descriptions. Finally, our evaluations primarily used policies with a CLIPort (Shridhar et al., 2021) architecture, but ASKDAGGER is not limited to this choice. It can be applied to any policy architecture where a teacher can determine the success of actions and provide demonstrations.

# 7 Conclusion

We have introduced the ASKDAGGER framework, which consists of three components: a gating procedure SAG that can track a user-specified metric; FIER for recasting novice actions to demonstrations; and PIER to prioritize replay experience based on uncertainty, novice success, and age. Our experiments show that ASKDAGGER allows for an effective balance between queries and failures based on a user-specified metric - sensitivity, specificity, or minimum system success rate. It significantly reduces the number of required annotated demonstrations while also improving its generalization capabilities to unseen scenarios by utilizing demonstrations acquired through interactive relabeling and validation. Additionally, it improves adaptation to domain shifts by prioritizing replay.

For future work, several promising directions emerge for further enhancing the capabilities of ASKDAGGER. Extending this methodology to scenarios involving longer horizons and non-sparse rewards could improve its applicability. Leveraging generative/foundation models and simulators for visualizing action plans and generating artificial demonstrations could also expand the application domain of ASKDAGGER, potentially improving training efficiency and outcome predictability. Additionally, applying ASKDAGGER to higher-frequency, low-level control tasks is another interesting future direction. Although relabeling and validation may become less beneficial with higher query frequencies, SAG could still effectively query human teachers when robot execution is slowed, combining robot-driven queries with human interventions. Integrating robot-gated ASKDAGGER for high-level affordance or skill learning with human-gated interventions for low-level control could create a hierarchical structure. This integrated approach would allow simultaneous on-policy demonstration collection across multiple control levels at different frequencies. Moreover, conducting participant studies would be vital to assess the mental load associated with ASKDAGGER and refine user interfaces accordingly. Finally, adapting ASKDAGGER to environments characterized by heterogeneous or imperfect teachers could widen its applicability and effectiveness in real-world settings.

# 8 Acknowledgments

Research reported in this work was partially or completely facilitated by computational resources and support of the Delft AI Cluster (DAIC) (Delft AI Cluster (DAIC), 2024) at TU Delft (RRID: SCR_025091), but remains the sole responsibility of the authors, not the DAIC team.

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

# A  Appendix

## A.1  Specificity-Aware Gating

For Specificity-Aware Gating, the threshold in line 12 of Alg. 2 is set as follows. The total specificity $\sigma^{\mathrm{spec}}$ can be computed by considering the true negatives $\mathrm{TN}_\gamma$ resulting from active queries, the false positives $\mathrm{FP}_{\mathrm{rand}}$ resulting from random queries, along with the false positives $\mathrm{FP}_\gamma$ from active querying:

$$\sigma^{\mathrm{spec}} = \frac{\mathrm{TN}_\gamma - \mathrm{FP}_{\mathrm{rand}}}{\mathrm{TN}_\gamma + \mathrm{FP}_\gamma}. \tag{7}$$

The number of $\mathrm{FP}_{\mathrm{rand}}$ are determined by which of $\mathrm{TN}_\gamma$ are queried. These queries occur with probability $p_{\mathrm{rand}}$ (see lines 8-9 of Alg. 1). Therefore, its expected value is $\mathrm{TN}_\gamma \cdot p_{\mathrm{rand}}$. This leads to:

$$\mathbb{E}_{\epsilon \sim U_{[0,1)}}[\sigma^{\mathrm{spec}}] = \frac{\mathrm{TN}_\gamma}{\mathrm{TN}_\gamma + \mathrm{FP}_\gamma} - \frac{\mathrm{TN}_\gamma \cdot p_{\mathrm{rand}}}{\mathrm{TN}_\gamma + \mathrm{FP}_\gamma} \tag{8}$$

$$= \sigma_\gamma^{\mathrm{spec}}(1 - p_{\mathrm{rand}}), \tag{9}$$

where $\sigma_\gamma^{\mathrm{spec}}$ is the specificity for gating disregarding the random queries. Thus, we interpolate between threshold values that best satisfy the desired specificity level $\sigma_{\mathrm{des}} = \sigma_\gamma^{\mathrm{spec}}(1 - p_{\mathrm{rand}})$ (line 12 of Alg. 2).

## A.2  Success-Aware Gating

The goal of Success-Aware Gating (SAG) is to minimize expert queries while maintaining a minimum system success rate. If the novice's success rate is below the desired success rate $\sigma_{\mathrm{des}}$, SAG increases expert queries to ensure that the combined success rate of expert and novice meets $\sigma_{\mathrm{des}}$. We assume that expert actions are always correct for this mode. If the novice's success rate is already at or above $\sigma_{\mathrm{des}}$, SAG does not actively query the expert to avoid redundant queries. The only case when the system fails is when the novice's action is invalid, while the expert is not actively queried and not randomly queried. Therefore, the system success rate is:

$$\sigma^{\mathrm{succ}} = 1 - \frac{\mathrm{FN}_\gamma - \mathrm{TP}_{\mathrm{rand}}}{\mathrm{TP}_\gamma + \mathrm{FP}_\gamma + \mathrm{TN}_\gamma + \mathrm{FN}_\gamma} \tag{10}$$

The number of $\mathrm{TP}_{\mathrm{rand}}$ are determined by which of $\mathrm{FN}_\gamma$ are queried. These queries occur with probability $p_{\mathrm{rand}}$ (see lines 8-9 of Alg. 1). Therefore, its expected value is $\mathrm{FN}_\gamma \cdot p_{\mathrm{rand}}$. This leads to:

$$\mathbb{E}_{\epsilon \sim U_{[0,1)}}[\sigma^{\mathrm{succ}}] = 1 - \frac{\mathrm{FN}_\gamma}{\mathrm{TP}_\gamma + \mathrm{FP}_\gamma + \mathrm{TN}_\gamma + \mathrm{FN}_\gamma} + \frac{\mathrm{FN}_\gamma \cdot p_{\mathrm{rand}}}{\mathrm{TP}_\gamma + \mathrm{FP}_\gamma + \mathrm{TN}_\gamma + \mathrm{FN}_\gamma} \tag{11}$$

$$= \sigma_\gamma^{\mathrm{succ}} + p_{\mathrm{rand}}(1 - \sigma_\gamma^{\mathrm{succ}}), \tag{12}$$

where $\sigma_\gamma^{\mathrm{succ}}$ is the system success rate disregarding the random queries. Thus, we interpolate between threshold values that best satisfy the desired system success rate $\sigma_{\mathrm{des}} = \sigma_\gamma^{\mathrm{succ}} + p_{\mathrm{rand}}(1 - \sigma_\gamma^{\mathrm{succ}})$ (line 12 of Alg. 2).

## A.3  SAG Ablations

SAG involves two regression steps, as shown in Alg. 2 and Fig. 2. The first step is linear regression to normalize $\boldsymbol{u}_W$, and the second is logistic regression to impute pseudo labels for cases without teacher feedback ($r = 0$). To evaluate the impact of these steps, we performed ablations in the same experimental setup as described in Sec. 5.1 (mode = sensitivity). We conducted dataset aggregation on the MNIST handwritten dataset under two conditions: one without normalization of $\boldsymbol{u}_W$ (w/o Normalization) and one without pseudo-label imputation (w/o Imputation). Fig. 12 and Tab. 1 compare these results to SAG with both normalization and imputation.

The sensitivity plots and table show a clear performance degradation for SAG w/o Imputation. Without imputing pseudo labels where success is unknown, the threshold is set too high, as gating results in labels

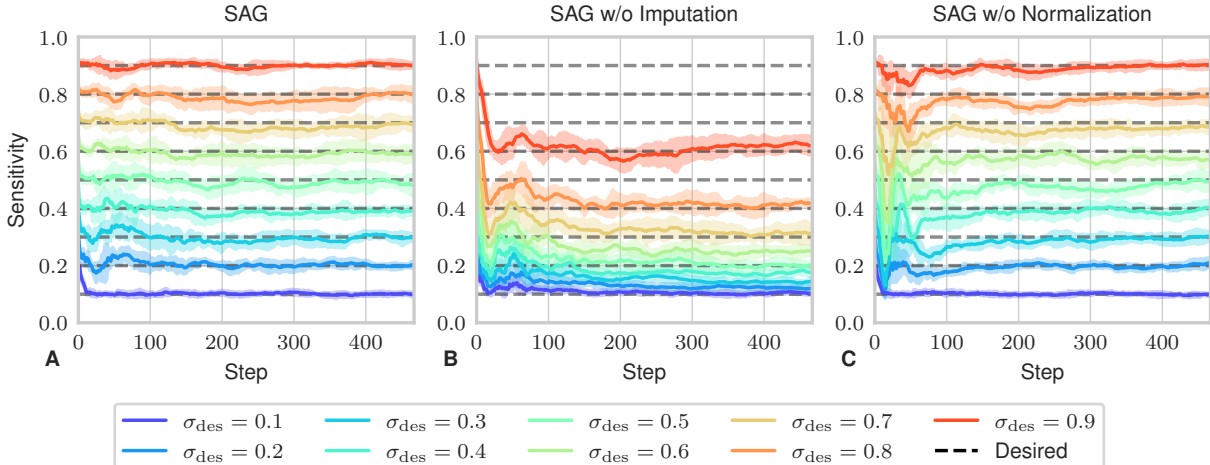

Figure 12: Comparison of SAG, SAG without pseudo-label imputation (w/o Imputation), and SAG without uncertainty normalization (w/o Normalization). Sensitivity is calculated over a moving window of 1000 failures. Mean and standard deviation are shown for ten repetitions. The plots show that SAG w/o Imputation sets the threshold too high, as it primarily uses labels from high-uncertainty regions. SAG w/o Normalization performs poorly in the early training stages due to rapidly decreasing uncertainty, which also results in an overly high threshold. A threshold that is too high will result in too low sensitivity, as not enough failures are prevented through querying.

Table 1: Comparison of SAG, SAG without pseudo-label imputation (w/o Imputation), and SAG without uncertainty normalization (w/o Normalization). Sensitivity is calculated over the complete training duration. Mean and standard deviation are shown for ten repetitions. Boldface is used to highlight the values corresponding to the best sensitivity tracking.

| $\sigma_{\text{des}}$ | SAG | SAG w/o Imputation | SAG w/o Normalization |
|---|---|---|---|
| 0.1 | $0.104 \pm 0.003$ | $0.114 \pm 0.003$ | $\mathbf{0.103 \pm 0.002}$ |
| 0.2 | $\mathbf{0.207 \pm 0.005}$ | $0.141 \pm 0.005$ | $0.190 \pm 0.004$ |
| 0.3 | $\mathbf{0.304 \pm 0.009}$ | $0.171 \pm 0.003$ | $0.272 \pm 0.006$ |
| 0.4 | $\mathbf{0.397 \pm 0.007}$ | $0.200 \pm 0.005$ | $0.365 \pm 0.010$ |
| 0.5 | $\mathbf{0.497 \pm 0.008}$ | $0.236 \pm 0.005$ | $0.448 \pm 0.010$ |
| 0.6 | $\mathbf{0.598 \pm 0.007}$ | $0.283 \pm 0.009$ | $0.548 \pm 0.012$ |
| 0.7 | $\mathbf{0.695 \pm 0.005}$ | $0.350 \pm 0.009$ | $0.655 \pm 0.013$ |
| 0.8 | $\mathbf{0.793 \pm 0.010}$ | $0.447 \pm 0.008$ | $0.764 \pm 0.005$ |
| 0.9 | $\mathbf{0.899 \pm 0.004}$ | $0.628 \pm 0.008$ | $0.880 \pm 0.007$ |

for only the high-uncertainty regions. This leads to overly low sensitivity values. Tab. 1 also indicates performance degradation when $\boldsymbol{u}_W$ is not normalized. Fig. 12 suggests that this primarily results from poor sensitivity tracking in the early training stages. Uncertainty drops rapidly during early training. Therefore, failing to normalize $\boldsymbol{u}_W$ causes the threshold to be set too high. As a result, sensitivity remains too low in these early stages.

We also performed an ablation to study the influence of the value of the random query probability $p_{\text{rand}}$. We compared the experiments from Sec. 5.1 with $p_{\text{rand}} = 0.1$ to training with $p_{\text{rand}} = 0.05$ and $p_{\text{rand}} = 0.2$. The results of this comparison are shown in Fig. 13, Tab. 2 and Tab. 3. The sensitivity plots in Fig. 13 A-C show that SAG can track multiple sensitivity levels for different values of $p_{\text{rand}}$. However, for $p_{\text{rand}} = 0.2$ and $\sigma_{\text{des}} = 0.1$ it fails. This results from SAG being unable to track sensitivities lower than $p_{\text{rand}}$. This follows

from Eq. (3):

$$\mathbb{E}_{\epsilon \sim U_{[0,1)}}[\sigma^{\text{sens}}] = \sigma_\gamma^{\text{sens}} + p_{\text{rand}}(1 - \sigma_\gamma^{\text{sens}}) \tag{13}$$

$$\geq p_{\text{rand}}, \tag{14}$$

since $0 \leq \sigma_\gamma^{\text{sens}} \leq 1$ and $0 \leq p_{\text{rand}} \leq 1$. The specificity plots in Fig. 13 D-F show that increasing $p_{\text{rand}}$ results in lower specificity values. This is also reflected in Tab. 3. This table shows the informedness values for the different values of $p_{\text{rand}}$. The informedness is also known as the Youden's J statistic and is calculated as:

$$\text{informedness} = \text{sensitivity} + \text{specificity} - 1. \tag{15}$$

It quantifies the performance of a dichotomous diagnostic test:

- below zero means worse than random;
- 0 equals random performance;
- Above zero indicates some degree of informed decision-making.

In Tab. 3, we see that increasing $p_{\text{rand}}$ results in lower informedness, as a higher percentage of queries results from random querying, rather than from querying based on exceeding the uncertainty threshold.

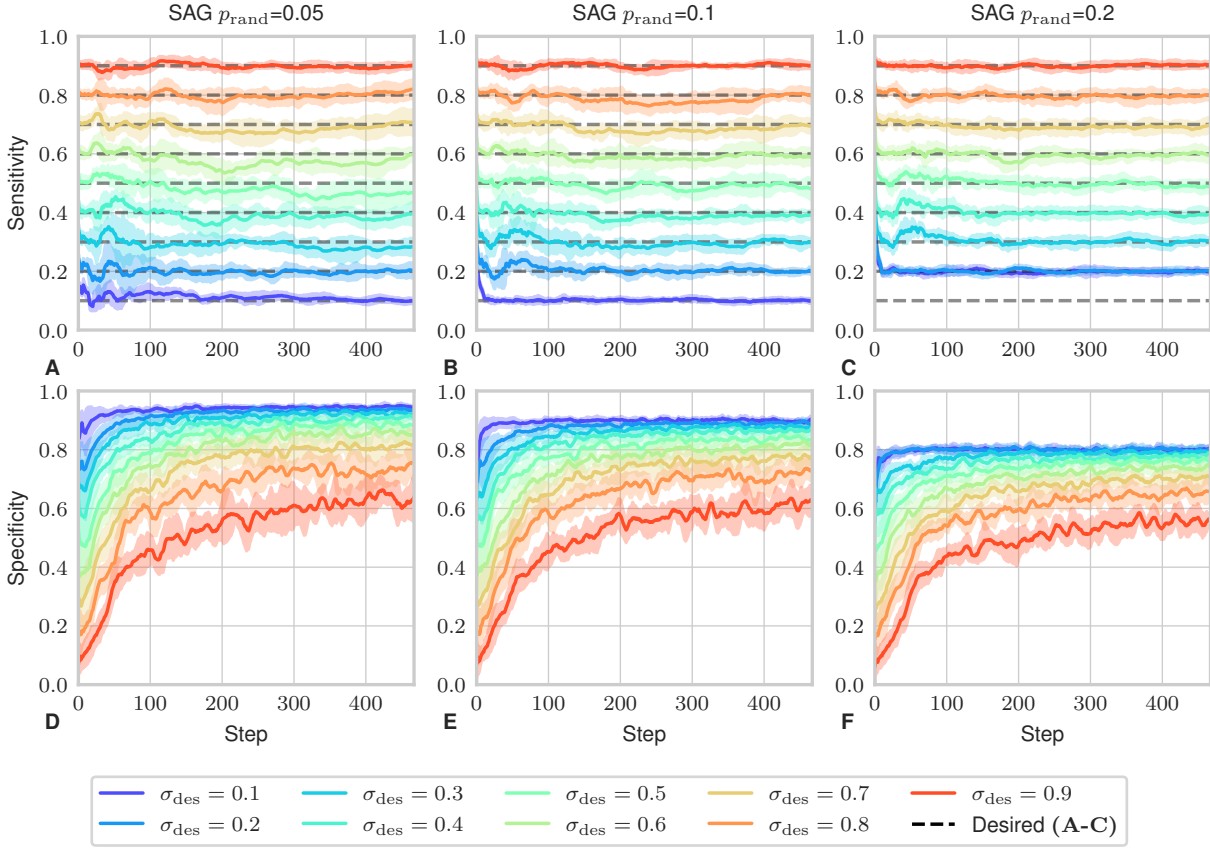

Figure 13: Comparison of SAG with different values of $p_{\text{rand}}$ (0.05 shown, 0.1 and 0.2). Sensitivity in **A-C** and specificity in **E-F** are calculated over a moving window of 1000 failures and successes, respectively. Mean and standard deviation are shown for ten repetitions. SAG can track a desired sensitivity for different values of $p_{\text{rand}}$. However, it fails for $\sigma_{\text{des}} = 0.1$ when $p_{\text{rand}} = 0.2$. This happens because the minimum trackable sensitivity is equal to $p_{\text{rand}}$. The plots **E-F** show that increasing $p_{\text{rand}}$ leads to lower specificity values.

Table 2: Comparison of SAG with different values of $p_{\mathrm{rand}}$. Sensitivity is calculated over the complete training duration. Mean and standard deviation are shown for ten repetitions. Boldface is used to highlight the values corresponding to the best sensitivity tracking.

| $\sigma_{\mathrm{des}}$ | sensitivity $p_{\mathrm{rand}} = 0.05$ | sensitivity $p_{\mathrm{rand}} = 0.1$ | sensitivity $p_{\mathrm{rand}} = 0.2$ |
|---|---|---|---|
| 0.1 | $0.113 \pm 0.006$ | $\mathbf{0.104 \pm 0.003}$ | $0.200 \pm 0.003$ |
| 0.2 | $\mathbf{0.198 \pm 0.012}$ | $0.207 \pm 0.005$ | $\mathbf{0.202 \pm 0.003}$ |
| 0.3 | $\mathbf{0.298 \pm 0.010}$ | $0.304 \pm 0.009$ | $0.309 \pm 0.005$ |
| 0.4 | $\mathbf{0.399 \pm 0.006}$ | $0.397 \pm 0.007$ | $0.406 \pm 0.007$ |
| 0.5 | $0.493 \pm 0.009$ | $\mathbf{0.497 \pm 0.008}$ | $0.507 \pm 0.007$ |
| 0.6 | $0.591 \pm 0.006$ | $0.598 \pm 0.007$ | $\mathbf{0.599 \pm 0.005}$ |
| 0.7 | $0.696 \pm 0.010$ | $0.695 \pm 0.005$ | $\mathbf{0.699 \pm 0.005}$ |
| 0.8 | $\mathbf{0.799 \pm 0.010}$ | $0.793 \pm 0.010$ | $\mathbf{0.799 \pm 0.006}$ |
| 0.9 | $0.897 \pm 0.003$ | $0.899 \pm 0.004$ | $\mathbf{0.900 \pm 0.006}$ |

Table 3: Comparison of SAG with different values of $p_{\mathrm{rand}}$. Informedness (Youden's J statistic = sensitivity + specificity $-1$) is calculated over the complete training duration. Mean and standard deviation are shown for ten repetitions. Boldface is used to highlight the values corresponding to the best informedness.

| $\sigma_{\mathrm{des}}$ | informedness $p_{\mathrm{rand}} = 0.05$ | informedness $p_{\mathrm{rand}} = 0.1$ | informedness $p_{\mathrm{rand}} = 0.2$ |
|---|---|---|---|
| 0.1 | $\mathbf{0.055 \pm 0.006}$ | $0.003 \pm 0.004$ | $-0.000 \pm 0.004$ |
| 0.2 | $\mathbf{0.126 \pm 0.011}$ | $0.092 \pm 0.004$ | $0.002 \pm 0.004$ |
| 0.3 | $\mathbf{0.209 \pm 0.012}$ | $0.170 \pm 0.010$ | $0.094 \pm 0.007$ |
| 0.4 | $\mathbf{0.284 \pm 0.007}$ | $0.245 \pm 0.007$ | $0.171 \pm 0.010$ |
| 0.5 | $\mathbf{0.349 \pm 0.013}$ | $0.311 \pm 0.015$ | $0.249 \pm 0.010$ |
| 0.6 | $\mathbf{0.410 \pm 0.010}$ | $0.375 \pm 0.013$ | $0.308 \pm 0.010$ |
| 0.7 | $\mathbf{0.464 \pm 0.011}$ | $0.427 \pm 0.014$ | $0.365 \pm 0.011$ |
| 0.8 | $\mathbf{0.483 \pm 0.025}$ | $0.460 \pm 0.019$ | $0.407 \pm 0.015$ |
| 0.9 | $\mathbf{0.456 \pm 0.027}$ | $0.445 \pm 0.014$ | $0.403 \pm 0.016$ |

### A.4 Prioritized Interactive Experience Replay (PIER) Intuition

Prioritized Interactive Experience Replay (PIER) prioritizes the replay of demonstrations based on novice success, uncertainty, and demonstration age. In Fig. 14, we visualize Eq. (6) to provide a more intuitive understanding of the prioritization scheme. The red lines correspond to novice failures ($r = -1$), while the green lines correspond to novice successes ($r = 1$). The black lines indicate $r = 0$, which occurs, for example, for relabeled demonstrations where the goal was relabeled, and we do not know whether the novice would act correctly. In this figure, we observe that for values of $b$ close to 1, the priorities converge almost linearly to 1, while for higher values of $b$, they converge exponentially.

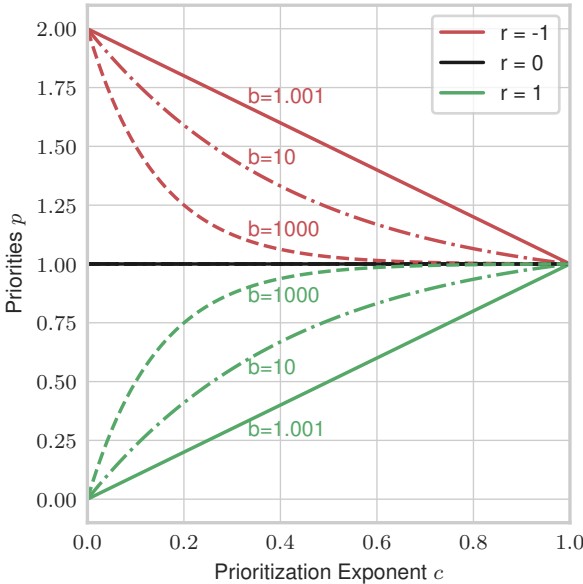

Figure 14: Visualization of Eq. (6) showing priorities $p$ over prioritization exponents $c$ for various values of base $b$.

In summary, we prioritize demonstrations in the following order (from highest to lowest):

1. Novice failure demonstrations that are recent and have low uncertainty.

2. Novice failure demonstrations that are either recent or have low uncertainty.

3. Novice failure demonstrations that are neither recent nor have low uncertainty.

4. Offline or relabeled demonstrations.

5. Novice success demonstrations that are neither recent nor have low uncertainty.

6. Novice success demonstrations that are either recent or have low uncertainty.

7. Novice success demonstrations that are both recent and have low uncertainty.

### A.5 Foresight Interactive Experience Replay (FIER) Oracle

The oracle used in the experiments from Sec. 5.2 is based on Shridhar et al. (2021). We extend this oracle to provide relabeling demonstrations. Instead of querying a human teacher about the validity of the novice's planned actions, we execute these actions in simulation to verify the plan and then reset the environment to its previous state.

### A.6 Considerations on Demonstration Types, Times, and Effort

The ASKDAGGER framework supports data collection through three modalities: annotation, relabeling, and validation. Each modality poses distinct challenges and places different demands on the user in terms of time and physical or cognitive effort. As in other imitation learning approaches in robotics, time efficiency is a more meaningful metric than data efficiency for evaluating real-world robot learning performance (Johns, 2022). A large portion of the overall data collection time is spent on environment setup, action execution, and resetting. We illustrate this with quantitative results from the experiments in Sec. 5.3. All reported timings are approximate and based on a 45-minute video recording of one experiment.

We measured the time required to provide each demonstration type in the assembly task. On average, validation took $3.7 \pm 1.9$ seconds, relabeling $7.5 \pm 1.9$ seconds, and annotation $7.0 \pm 3.8$ seconds. Based on these values, one might conclude that relabeling is the most time-consuming, while validation is the fastest. However, following Johns (2022), a time-efficiency analysis should also account for the full process involved in producing a demonstration, including environment resetting, issuing robot commands, model inference, and action execution. In our setup: Resetting the environment took 8 seconds and was required every four actions, averaging 2 seconds per action. Command generation was automated and took about 1 second to execute. CLIPort model inference (including sensor processing, preprocessing, and interface delays) took roughly 5 seconds. Action execution took approximately 25 seconds. Accounting for these overheads, a validation demonstration takes about 37 seconds in total, and an annotation demonstration takes about 40 seconds. In contrast, a relabeling demonstration can be provided when prompted during an annotation query and does not require additional resets, executions, or inference. Thus, its effective cost remains close to 8 seconds, making it the most time-efficient modality overall. Moreover, since relabeled demonstrations are not executed, relabeling does not block robot execution and can be done in parallel. We emphasize that these timings are highly task- and interface-dependent and are provided here as illustrative examples. Furthermore, time is only one aspect of demonstration cost; cognitive load and mental effort of the human demonstrator are also important factors.

As noted, the cost (teaching effort) of each demonstration type is task- and interface-dependent. Nonetheless, we can consider different scenarios for the experiments described in Sec. 5.2 and evaluate them. We discuss three such scenarios (these are not exhaustive and may differ in likelihood):

- **Scenario 1**: Annotation demonstrations dominate the demonstration cost.

- **Scenario 2**: Annotation and validation demonstrations are equally costly and dominate the overall cost.

- **Scenario 3**: All demonstration types are equally costly.

Results for these scenarios are shown in Fig. 15. In Scenario 1, ASKDAGGER shows a clear advantage over other methods, achieving higher success rates during training for the same number of annotation demonstrations. In Scenario 2, performance differences during training are less pronounced. However, ASKDAGGER still outperforms other methods on the *unseen* tasks, as shown in Fig. 5. In Scenario 3, where all demonstration types incur the same cost, ASKDAGGER has the highest total demonstration cost during training. Nonetheless, since it still generalizes better to *unseen* tasks, whether to collect relabeling demonstrations becomes a trade-off between generalization performance and demonstration cost.

### A.7 CLIPort Implementation

Our implementation of CLIPort agents builds on Shridhar et al. (2021) with modifications for interactive imitation learning. Efficient model updates are essential in this setting, as saving multiple checkpoints and performing evaluation rollouts after each update is impractical. Thus, training stability is a priority. We replace rectified linear units (ReLUs) with leaky ReLUs to prevent vanishing gradients. We also use a larger batch size (8 in simulation, 3 in real-world experiments) than the original (1) (Shridhar et al., 2021). Our batch training implementation is based on Wu (2024). Since batch training increases memory requirements, we reduce the model size by removing the two middle layers from the ResNet streams with lateral connections, the first language fusion layer, and its associated upscaling and lateral fusion layers.

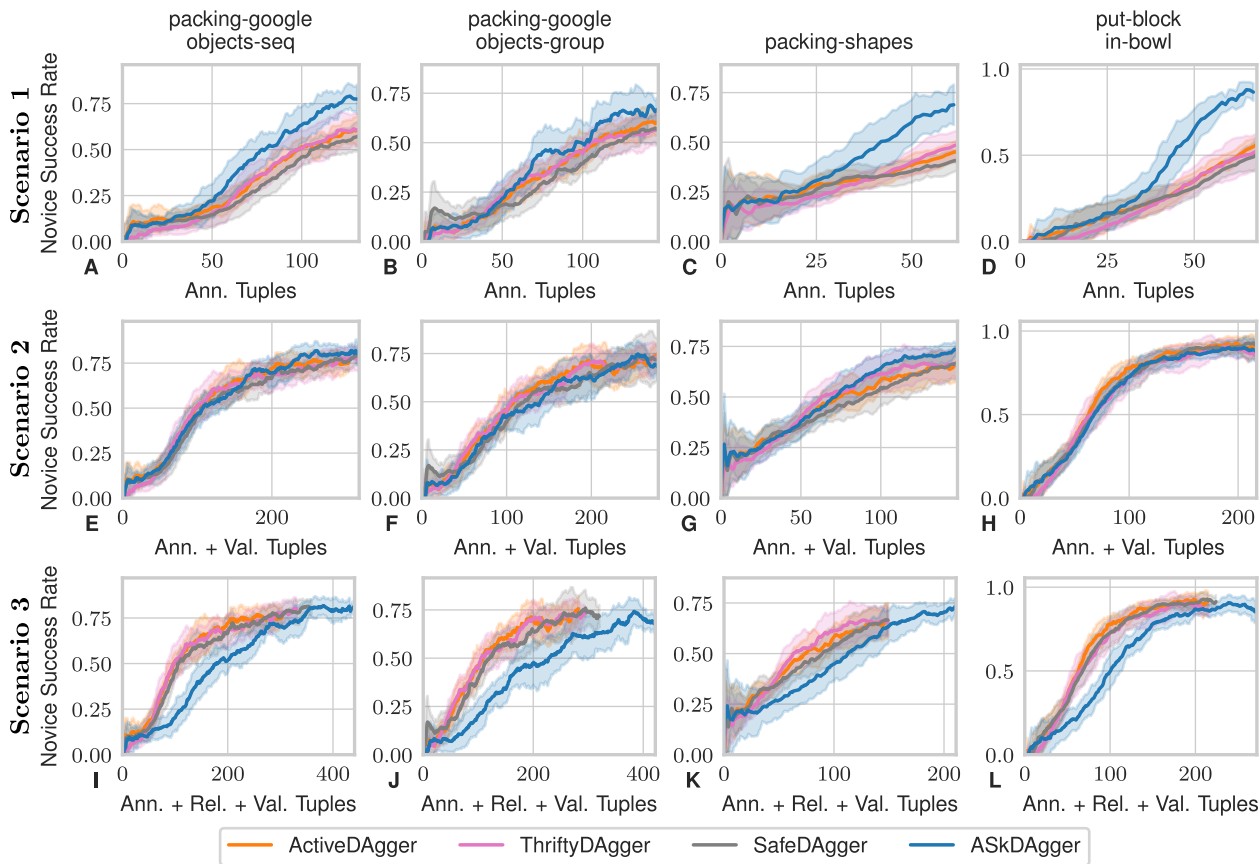

Figure 15: Novice success rate during training as a function of the number of collected annotation tuples (top row), annotation + validation tuples (middle row), and annotation + validation + relabeling tuples (bottom row). Mean and standard deviation are computed over ten policies using a moving window of 50 episodes. The top row shows that when annotation demonstrations are the dominant demonstration cost (scenario 1), ASKDAGGER achieves a clear performance gain on both *seen* and *unseen* tasks, benefiting from validation and relabeling demonstrations. The middle row shows that when annotation and validation demonstrations are equally costly and dominate the total cost (scenario 2), all methods perform similarly relative to demonstration cost, with ASKDAGGER offering a performance gain only on *unseen* tasks. Finally, the bottom row shows that when all demonstration types are equally costly (scenario 3), ASKDAGGER performs better on *unseen* tasks but requires more costly training on *seen* tasks.

## A.8 CLIPort Tasks

The CLIPort (Shridhar et al., 2021) benchmark defines *seen* and *unseen* settings. In the *seen* setting, commands are identical to those used during training. In the *unseen* setting, test-time commands involve different objects, shapes, or colors. We use the same *seen* and *unseen* sets as in the experiments by Shridhar et al. (2021). The only difference is that, for tasks involving shapes and Google objects, we include shapes and objects from the *unseen* set as distractors during training. Tab. 4 provides a complete overview of the *seen* and *unseen* sets.

## A.9 THRIFTYDAGGER Implementation

Similar to Hoque et al. (2022), gating is based on novelty and riskiness in our implementations of ThriftyDAgger. For our novelty measure, we take the mean entropy of the pixel-wise value estimates over 5 dropout evaluations with a dropout rate of 0.3. We define riskiness as one minus the maximum value of the pixel-wise value estimates during evaluation without dropouts.

Table 4: Specifications of the *seen* and *unseen* sets in the CLIPort benchmark tasks (Shridhar et al., 2021). Shapes are from Zeng et al. (2021), and Google scanned objects are from Downs et al. (2022).

|  | Seen | Unseen | All |
|---|---|---|---|
| Colors | cyan, yellow, brown, gray | pink, orange, purple, white | green, red, blue |
| Shapes | letter R shape, letter A shape, triangle, square, plus, letter T shape, diamond, pentagon, rectangle, flower, star, circle, letter G shape, letter V shape | letter E shape, letter L shape, ring, hexagon, heart, letter M shape | |
| Google Scanned Objects | alarm clock, android toy, black boot with leopard print, black fedora, black razer mouse, black sandal, black shoe with orange stripes, bull figure, butterfinger chocolate, c clamp, can opener, crayon box, dog statue, frypan, green and white striped towel, grey soccer shoe with cleats, hard drive, honey dipper, magnifying glass, mario figure, nintendo 3ds, nintendo cartridge, office depot box, orca plush toy, pepsi gold caffeine free box, pepsi wild cherry box, porcelain cup, purple tape, red and white flashlight, rhino figure, rocket racoon figure, scissors, silver tape, spatula with purple head, spiderman figure, tablet, toy school bus | ball puzzle, black and blue sneakers, black shoe with green stripes, brown fedora, dinosaur figure, hammer, light brown boot with golden laces, lion figure, pepsi max box, pepsi next box, porcelain salad plate, porcelain spoon, red and white striped towel, red cup, screwdriver, toy train, unicorn toy, white razer mouse, yoshi figure | |

## A.10 SAFEDAGGER Implementation

In our experiments, SafeDAgger (Zhang & Cho, 2017) is implemented as follows. In the original formulation of SafeDAgger, a safety classifier is learned that takes as input the observation and novice policy and outputs a binary prediction of whether the novice will deviate from the teacher policy. Later, more general definitions of SafeDAgger have been introduced, e.g., by Menda et al. (2019), who formalize SafeDAgger as a decision rule based on a discrepancy measure between novice and teacher actions. In our implementation, the decision rule is based on the Q-value of the pixel-wise value estimates of the CLIPort model, since Zhang & Cho (2017) describe that the safety policy is similar to a value function in the case of sparse rewards.

## A.11 CLIPort Ablation

The comparison of ASKDAGGER with the baselines in Fig. 4 shows a clear performance improvement on the *unseen* tasks. We attribute this gain to the recasting of failures as demonstrations, enabled by FIER (claim **C3**). To support this explanation, we conducted an ablation study where ASKDAGGER is run without relabeling (ASKDAGGER w/o relabeling), the only change being the omission of relabeling. The results,

shown in Fig. 16, confirm that the performance improvement on *unseen* tasks is indeed due to relabeling, thus supporting claim **C3**. With relabeling, ASKDAGGER achieves an average evaluation reward improvement of 62% across the four *unseen* tasks.

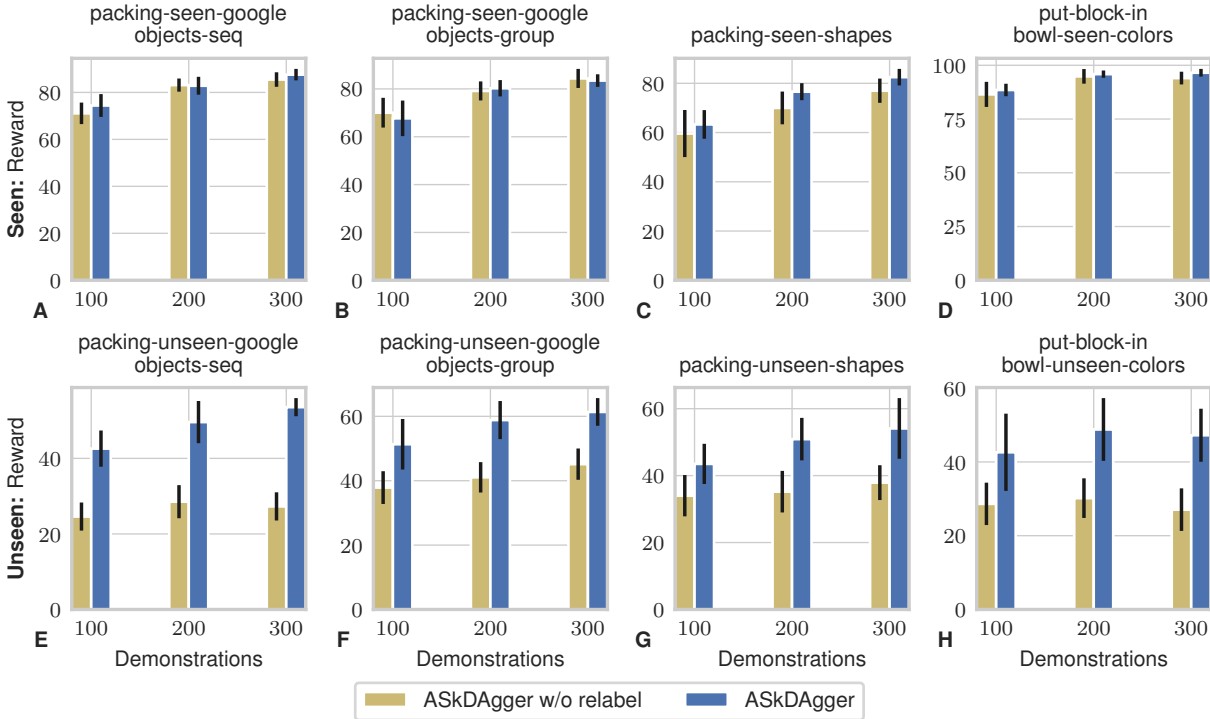

Figure 16: Cumulative rewards for evaluating checkpoints over 100 episodes on tasks with *seen* and *unseen* objects. Mean and standard deviation are shown for ten policies using a moving window of 50 episodes. The results show that relabeling demonstrations improve the performance of ASKDAGGER on the *unseen* scenarios.

## A.12 Comparison between ASkDAgger and Affordance Learning with Where2Act

In this section, we highlight the difference between ASKDAGGER, a data aggregation strategy, and affordance learning methods (Mo et al., 2021; Mazzaglia et al., 2024; Wang et al., 2022; Ning et al., 2023; Geng et al., 2023), which are designed to efficiently explore the large state-action space of affordances. ASKDAGGER collects on-policy, goal-conditioned data and is tailored to perform well on the distribution of states and goals encountered by the novice policy. In contrast, affordance learning methods such as Where2Act (Mo et al., 2021) aim not to provide accurate predictions for a specific state distribution induced by a policy, but to cover a broad range of the state-action space and typically require constant reward information. In short, ASKDAGGER optimizes for on-policy performance, while affordance learning methods like Where2Act prioritize coverage and generalization. To illustrate these differing objectives, we compare ASKDAGGER with Where2Act on a task where we finetune a Where2Act model for the skill pushing, trained on the 10 training categories from Mo et al. (2021), to the single category *StorageFurniture* (see Fig. 18). For comparison, ASKDAGGER uses the same model architecture as Where2Act, and we do not collect expert demonstrations of successful poses or relabeling demonstrations. As we use the Where2Act model for the novice policy, we can also learn from failures and aggregate those into the dataset as well. Its policy is based on the prediction-biased adaptive data sampling strategy from Where2Act.

Therefore, the main differences between the two methods in this setting are: ASKDAGGER uses the PIER replay prioritization method (while Where2Act samples successes and failures with equal probability), ASKDAGGER includes the SAG gating mechanism, and Where2Act explores using random actions 50% of the time.

We used the following settings and parameters. For both methods, we collect data from the same number of interactions (800) with the same update-to-data ratio (4 update steps every 16 interactions). For ASKDAGGER, we use the sensitivity mode, with $\sigma_{\text{des}} = 0.9$. For ASKDAGGER, we quantify the uncertainty $u$ based on a least confidence approach (Settles, 2009) based on the state-action value function $Q$, i.e. $u = |\mathbb{1}\{Q(\boldsymbol{o}, \boldsymbol{a}) > 0.5\} - Q(\boldsymbol{o}, \boldsymbol{a})|$. For PIER, we used $\alpha = 0.5$, $\lambda = 0.5$, $b = 10$, and $\beta = 0$. Please note that these parameters were not tuned.

The results for these experiments are shown in Fig. 17. These plots clearly show the different objectives of ASKDAGGER and Where2Act. While ASKDAGGER performs better on-policy and on the *seen* shapes, Where2Act explores more and performs better on the *unseen* shapes.

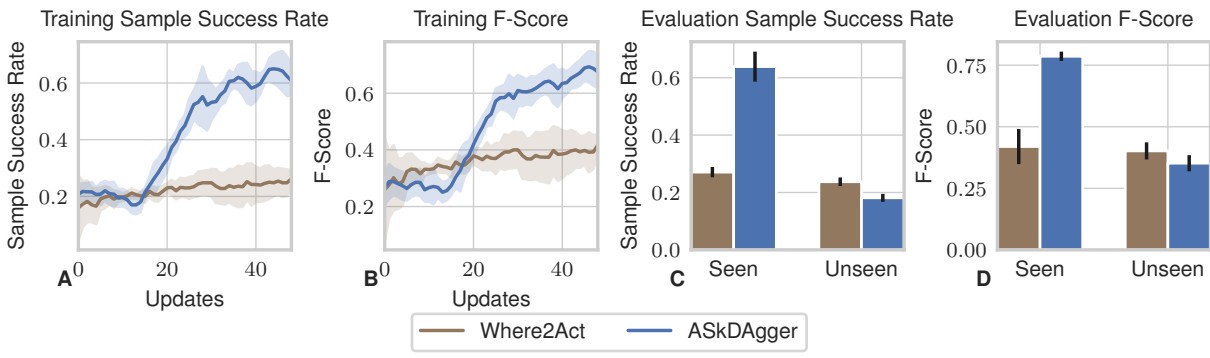

Figure 17: Training and evaluation sample success rates and F-scores (balance between precision and recall) are shown for finetuning on a single object category (*StorageFurniture*) for a pushing skill. We also show evaluation results on *seen* an *unseen StorageFurniture* shapes. Mean and standard deviation are shown for five repetitions. Random actions from Where2Act are excluded when calculating sample success rates and F-scores during training. For comparison, ASKDAGGER was trained without annotation or relabeled demonstrations. ASKDAGGER is designed for on-policy data collection and performs better on the *seen* set of shapes. On the other hand, Where2Act (Mo et al., 2021) is designed to efficiently explore the state-action space, leading to better generalization to *unseen* shapes.

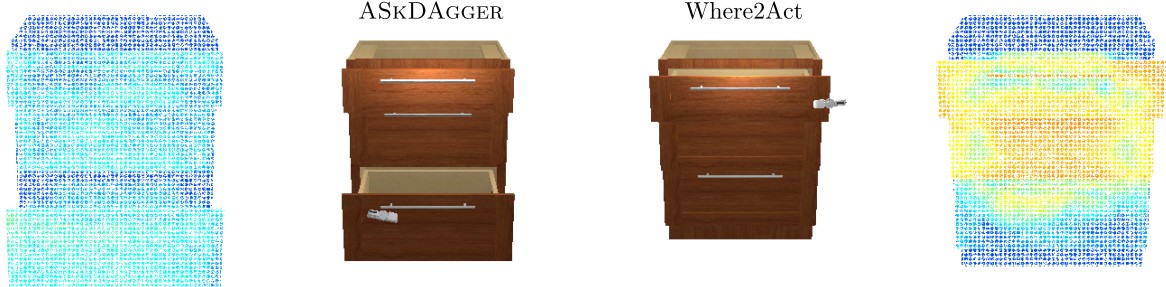

Figure 18: Visualization of value maps and high-value proposals from ASKDAGGER (left) and Where2Act (Mo et al., 2021) (right). Results are shown after finetuning on a single object category for a pushing skill and evaluating on an *unseen* shape from that category. ASKDAGGER is designed for on-policy data collection and performs better during policy execution. On the other hand, Where2Act (Mo et al., 2021) performs active exploration, leading to better generalization to *unseen* shapes, as qualitatively shown in the figure.

## A.13 Compute Resources

The MNIST experiments (Sec. 5.1) and real-world experiments (subsection 5.3 and subsection 5.4) were performed using an RTX 3080 Mobile graphics card. The CLIPort simulation benchmark experiments (Sec. 5.2) were performed using multiple A40 graphics cards on a high-performance computing cluster.

