# OpenReview forum: "ASkDAgger: Active Skill-level Data Aggregation for Interactive Imitation Learning"
_TMLR — Accepted by TMLR_

### Review · Reviewer_1FdY · 2025-05-18

**Summary Of Contributions:**

This paper proposes an enhanced Interactive Imitation Learning (IIL) framework building on DAgger. Unlike existing active DAgger approaches that transfer control to the human teacher during queries, their method enables the novice to also communicate its intended action when uncertain. This allows the teacher to either validate or correct the novice's plan, offering richer and more targeted feedback. They present three key algorithmic contributions addressing: (1) when to initiate interaction, (2) how to structure the interaction, and (3) how to effectively utilize the resulting demonstrations.

**Audience:**

Yes

**Claims And Evidence:**

Yes

**Requested Changes:**

Please check the weaknesses.

**Strengths And Weaknesses:**

### Strengths

1. The research addresses an important problem, particularly in tackling the challenges of data inefficiency and poor generalization in robotic learning.

2. The algorithm is presented clearly and in a well-structured manner.

3. The empirical validation includes both simulation and real-world robotic experiments, enhancing the credibility of the results.

4. The performance on unseen tasks is impressive, demonstrating strong generalization capabilities.

### Weaknesses

1. The research gap relative to existing work is not clearly articulated, which weakens the motivation behind the algorithmic design.

2. The three proposed components are largely heuristic in nature—for instance, using thresholds for uncertainty estimation and assigning weights for prioritized replay. As a result, the novelty of the approach appears limited.

3. The baseline comparisons are primarily ablated versions of the proposed method, with only one external baseline (Active DAgger). Including a broader set of baselines would strengthen the evaluation.

4. It remains unclear why the proposed algorithm should lead to improved generalization. A deeper theoretical or empirical justification would be beneficial.

---

> ### Author Response · Authors · 2025-06-29
> **Response to Reviewer 1FdY**
>
> Dear Reviewer 1FdY,
>
> First, we would like to thank you for your review. We have revised the manuscript accordingly, with changes highlighted in blue.
>
> =====Articulation of the Research Gap=====
>
> We have addressed your comment related to the articulation of the research gap by extending the related work section (section 2) to include related papers from other learning paradigms as well, and by adding a paragraph that states the research gap explicitly.
>
> =====Novelty of the Components=====
>
> Our method comprises three components: SAG for gating based on sensitivity, specificity, or success rate; FIER for collecting demonstrations through validation and relabeling of novel actions; and PIER for replay prioritization based on uncertainty, novice success, and demonstration age. We emphasize that these components are not independent. FIER gathers reward information for queried actions, which is used by both SAG and PIER. SAG relies on this information to relate uncertainty estimates to outcomes and adjust thresholds adaptively, while PIER uses both uncertainty and reward signals to guide replay prioritization. This interdependence forms a cohesive feedback loop that links uncertainty quantification, demonstration collection, and demonstration replay. The novelty of AIDA lies not only in each individual module but also in their integration into a unified framework.
>
> While relabeling with Hindsight Experience Replay (HER) [1] and replay prioritization with Prioritized Experience Replay (PER) [2] are established within the field of reinforcement learning, equivalent mechanisms have not been developed for IIL. Our work introduces PIER and FIER, which adapt these ideas to the IIL setting where rewards are not available. These components improve generalization and adaptation to novel scenarios, as shown in our experiments. This highlights that their introduction brings both novelty and practical value to IIL.
>
> We also introduce SAG as a novel mechanism for regulating gating in IIL. It supports user-specified targets for sensitivity, specificity, or success rate, enabling users to tailor gating based on task-specific requirements. SAG adaptively sets uncertainty thresholds at each time step to meet the desired metric, eliminating the need for heuristic threshold selection by the user. This approach generalizes beyond fixed-threshold methods by allowing explicit control over key behavioral properties such as safety, query efficiency, and task success. Importantly, estimating these metrics is non-trivial, since feedback is only available for queried actions, as the robot operates autonomously the rest of the time without supervision. To overcome the problem of estimating and tracking these metrics, we propose a pseudolabeling strategy using logistic regression to estimate reward outcomes for non-queried actions. This improves the estimation of gating metrics and enhances the performance of SAG as it mitigates the bias caused by querying high-uncertainty actions only. Ablation results in Appendix A.3 show that this extension leads to substantial performance gains in tracking the desired metric.
>
> =====Baseline Comparisons=====
>
> We thank the reviewer for pointing out that more baseline comparisons would strengthen the evaluation. We have addressed this by including two additional external methods in section 5.2: SafeDAgger [3] and ThriftyDAgger [4]. These methods are well-known active DAgger variants, making them relevant baselines for comparison. Our results show that AIDA requires fewer annotated demonstrations while achieving better performance on unseen tasks, primarily due to the relabeling mechanism introduced by FIER.
>
> =====Improved Generalization=====
>
> We thank the reviewer for highlighting the need to clarify the generalization behavior of our method to unseen scenarios. Accordingly, we have clarified the task setup in Section 5.2.
>
> Furthermore, we conducted additional ablations to provide stronger empirical support for our generalization claims. In particular, we added a comparison between AIDA and AIDA without relabeling to isolate the effect of relabeled demonstrations. The results, presented in Appendix A.8, confirm that the improved performance on unseen tasks is indeed due to the inclusion of relabeled demonstrations. They show an average performance gain of 62% thanks relabeling demonstrations with FIER on the unseen tasks.
>
> References
>
> [1] Andrychowicz, M., Wolski, F., Ray, A., Schneider, J., Fong, R., Welinder, P., ... & Zaremba, W. Hindsight experience replay. NeurIPS. 2017
>
> [2] Schaul, T., Quan, J., Antonoglou, I., & Silver, D. Prioritized experience replay. arXiv. 2015
>
> [3] Zhang, J., & Cho, K. Query-efficient imitation learning for end-to-end simulated driving. AAAI. 2017
>
> [4] Hoque, R., Balakrishna, A., Novoseller, E., Wilcox, A., Brown, D. S., & Goldberg, K. ThriftyDAgger: Budget-Aware Novelty and Risk Gating for Interactive Imitation Learning. CoRL. 2022

---

### Review · Reviewer_Uc41 · 2025-05-19

**Summary Of Contributions:**

AIDA (Action Inquiry DAgger) is a novel IIL framework that allows a novice policy to communicate its intended action alongside its uncertainty to a teacher policy. The authors introduce the following three modules in their IIL framework:
1. Sensitivity-Aware Gating (SAG): a mechanism to dynamically adjust the threshold for querying the teacher based on a user-defined sensitivity level (true positive rate).
2. Foresight Interactive Experience Replay (FIER): for aggregating valid and relabeled novice action plans into the demonstration dataset.
3. Prioritized Interactive Experience Replay (PIER): for prioritizing replay based on uncertainty, novice success, and demonstration age.

Authors show experiments demonstrating the efficacy of each component, through learning a classifier on MNIST, keyframe actions on CLIPort tasks, and two real-world sorting tasks. Through their experiments, the authors showcase their different proposed components work and help train decision-making models with greater sample efficiency.

**Audience:**

Yes

**Broader Impact Concerns:**

No concern.

**Claims And Evidence:**

Yes

**Requested Changes:**

Please see weaknesses in the above section. In summary, below are the requested changes for strengthening the paper in my opinion:
1. More clarity about action space in problem setting.
2. More clarity around CLIPort experiments - description of the input and output of the model, details on generalization experiments, and discussion on technical contribution to object generalization.

**Strengths And Weaknesses:**

**Strengths:**
1. I find the paper well-written and easy to follow. The authors made it very clear in the introduction what the scope and key contributions of this paper are.
2. The paper shows strong results in support of its claims. The MNIST experiment is specifically designed to test whether SAG can accurately track and maintain a desired sensitivity level over time. CLIPort tasks involve language-conditioned, vision-based robotic manipulation representing complex, real-world challenges such as manipulation involving unseen objects.

**Weaknesses:**
1. For a large portion of this paper, it is unclear what is the application space is for this paper esp. in robotics. While the authors eventually make it clear that their action space is the space of mid-to-high level actions, one could be confused if this method was trying to improve an end-to-end visuomotor policy, which it isn’t. I think that point can be made clearer in the introduction or in Section 3 what the exact output/action spaces are in the experiments.
2. In the CLIPort experiments (Section 5.2), it is unclear what a dataset entry is (input and output of the model). Is it an image and two 6-DoF actions? What is the space of language instructions that can be provided for each table-top setup?
3. In the CLIPort experiments, what is the space of “unseen” objects? Can the authors explain why AIDA generalizes better than the baseline on unseen objects. As far as I understand, no technical contribution directly helps achieve better generalization.

---

> ### Author Response · Authors · 2025-06-29
> **Response to Reviewer Uc41**
>
> Dear Reviewer Uc41,
>
> First, we would like to thank you for your constructive review. We have revised the manuscript accordingly, with all changes highlighted in blue. Below, we respond to your comments in detail.
>
> =====Application Space of AIDA=====
>
> We agree that the application space of AIDA should be stated more clearly. To address this, we have revised the introduction to include the following clarification:
>
> “Since AIDA relies on the novice communicating its planned actions for teacher feedback, it is suited to mid- to high-level control tasks rather than end-to-end policy learning. It is most applicable in scenarios where a robot has access to predefined skills such as grasping, walking, pushing, door opening, screwing, or inserting.”
>
> We have also updated the problem statement section (Sec. 3) to specify the nature of the action space:
>
> “We focus on mid- to high-level control tasks, where the novice has access to a set of predefined skills such as walking, grasping, or inserting. The novice learns from a demonstration dataset consisting of trajectories. These trajectories consist of the parameters of the demonstrated skills provided by the teacher.”
>
> =====Clarity around CLIPort Experiments=====
>
> We appreciate the suggestion to improve the clarity and self-contained nature of the CLIPort experiment descriptions. We have revised Section 5.2 (CLIPort experiments) to specify the model inputs, outputs, and dataset format as follows:
>
> “The observations in these experiments consist of an RGB-D image and a natural language command. CLIPort employs a two-stream architecture: a spatial stream and a semantic stream. The semantic stream uses frozen CLIP encoders [2] to extract features from the RGB image and the language command. The spatial stream is an untrained Transporter network [3], whose decoder layers are fused with features from the semantic stream. The model outputs pixel-wise value estimates for both picking and placing. The demonstration dataset includes RGB-D images, language commands, and expert actions in the form of Cartesian pick and place poses.”
>
> We have also clarified the setup for generalization to unseen objects:
>
> “The CLIPort benchmark includes seen and unseen task settings. In the unseen setting, test-time
> commands involve different objects, shapes, or colors than during training. For example, the set of seen colors = \{yellow, brown, gray, cyan\} and the set of unseen colors = \{purple, pink, white, black\}. Some colors appear in both settings, i.e., the set of all colors = \{red, green, blue\}. A complete list of seen and unseen objects is provided in Fig. 16 in App. A.5. We modified the tasks involving Google objects and shapes by sometimes introducing unseen objects as distractors during training, as real-world scenarios also involve varying distractor objects. Therefore, it is possible for AIDA to acquire demonstrations for the unseen set via relabeling with FIER when the novice fails in a meaningful way. This setup allows us to provide evidence for claim C3 by evaluating whether failures can be relabeled and whether this improves generalization to the unseen setting. ''
>
> To summarize, FIER converts meaningful failures into training data, expanding the effective training distribution. Moreover, the use of pretrained (foundation) models may increase the likelihood of meaningful failures, as these models can possibly extract relevant features across a wide range of objects. In our experiments, relabeling with FIER results in an average reward improvement of 62% in the unseen tasks without additional expert annotation demonstrations.
> These clarifications directly support claim C3 and address your concern regarding generalization performance.
>
> References:
>
> [1] M. Shridhar, L. Manuelli, and D. Fox. CLIPort: What and where pathways for robotic manipulation. In Proc. of the Conf. on Robot Learning (CoRL), 2021.
>
> [2] A. Radford, J. W. Kim, C. Hallacy, A. Ramesh, G. Goh, S. Agarwal, G. Sastry, A. Askell, P. Mishkin, J. Clark, et al. Learning transferable visual models from natural language supervision. In Proc. of the Intl. Conf. on Machine Learning (ICML), pp. 8748–8763. PMLR, 2021
>
> [3] A. Zeng, P. Florence, J. Tompson, S. Welker, J. Chien, M. Attarian, T. Armstrong, I. Krasin, D. Duong, V. Sindhwani, et al. Transporter networks: Rearranging the visual world for robotic manipulation. In Proc. of the Conf. on Robot Learning (CoRL), pp. 726–747. PMLR, 2021.

---

### Review · Reviewer_C8RH · 2025-06-04

**Summary Of Contributions:**

The work presents AIDA (Action Inquiry DAgger), a framework for interactive imitation learning (IIL).
AIDA aims to reduce human effort and improve performance in IIL by introducing three major distinctive features:
* SAG (Sensitivity-Aware Gating): which tracks a sensitivity level to decide when it's necessary to query the demonstrator(s) for information;
* FIER (Foresight Interactive Experience Replay): expands the demonstrator-provided information with trajectory relabeling and validation of the action plans from the IIL novice;
* PIER (Prioritized Interactive Experience Replay): biases the sampling process from the replay buffer based on uncertainty, novice success and demonstration age.
AIDA is evaluated in toy settings, and simulated and real-world high-level actions robotic setups, using CLIPort [1].

[1] CLIPort: What and Where Pathways for Robotic Manipulation, Shridhar et al, 2021

**Audience:**

Yes

**Broader Impact Concerns:**

The work presents no ethical risks. Potential misuse of robotic systems in real-world can be unsafe, if no safeguard mechanisms are applied to the system and the hardware. Adding a short statement in the paper would be appropriate.

**Claims And Evidence:**

Yes

**Requested Changes:**

### Major
* **Details about the relabeling and validation time and effort**: see related Weakness
* **Limited applicability?**: it is unclear to me why the method is not applicable in higher-frequency control problems. While it is generally more convenient to provide a feedback or an annotation on higher-level problem, it should be possible to apply AIDA also on more complex low-level tasks. For instance, modern teleoperation frameworks allow humans to intervene and provide demonstrations when the agent requires it.
* **Comparison with affordance-based / reinforcement learning methods**: with the addition of SAG and FIER, the method resembles affordance-based methods using online learning techniques, such as the adaptive sampling in [5] and the information-driven sampling in [6]. Compared to IIL methods, these approaches learn better from failures, relying on supervised or reinforcement learning. A comparison with such methods could be a nice way to expand the paper beyond the CLIPort policy learning setup. Alternatively, I would recommend the authors to discuss more in details what differentiates their method from affordance supervised learning / reinforcement learning / goal-conditioned reinforcement learning techniques.
### Minor
* It would be interesting to see what's the performance of a Dagger (or Active Dagger) approach when the number of annotations is similar to the sum of (annotations + relabeled + validation) for AIDA
* Figure 1 is quite generic and not particularly helpful to understand the method better at the moment. It would be best if they authors provided a more detailed illustration of the method(s) presented

[4] Teleoperation with Immersive Active Visual Feedback, Cheng et al, 2024

[5] Where2Act: From Pixels to Actions for Articulated 3D Objects, Mo et al, 2021

[6] Information-driven Affordance Discovery for Efficient Robotic Manipulation, Mazzaglia et al, 2024

**Strengths And Weaknesses:**

## Strengths:
* **Clear writing**: the paper clearly contextualizes the problem and states the claims to be verified in the Introduction. Then, it proceeds to effectively present the problem settings and method. Finally, it presents the experimental results while recalling the claims from the introduction. Overall, the flow is linear and the work is easy to follow.
* **Novelty**: FIER and PIER techniques are heavily inspired by techniques often applied in Reinforcement Learning (HER [2] and PER [3], respectively). However, their application is less common in the IIL context and the authors fit them elegantly within their framework. The idea behind SAG also contains novelty, despite uncertainty quantification methods being common for active learning in robotics.
* **Open**: the authors released code and assets for their work, demonstrating high-reproducibility and transparency in their work

## Weaknesses:
* **Complexity**: the introduced components make the method more complex and relying on different design choices, such as the method to estimate uncertainty in SAG, and additional hyperparameters for SAG and PIER. E.g. the probability of randomly sampling a query and the desired sensitivity in SAG, sampling hyperparameters in PIER. Nonetheless, the authors already ablate most of these components and provide at least an explanation on how they should be selected.
* **Demonstrator's effort**: it is not clear whether the labelling methods introduced in FIER reduce the workload of the demonstrator. The authors only compare the number of annotations, e.g. in Figure 6 and 11. However, it would be advisable to quantify if there's any time or effort saved by reducing the number of annotations, in favor of relabeling and validation.
* **Evaluation focussed on CLIPort**: given the multiple robotics experiments presented, I would have expected at least one of them to not be heavily relying on CLIPort for policy learning. See proposed changes for further details.


[2] Hindsight Experience Replay, Andrychowicz et al, 2017

[3] Prioritized Experience Replay, Schaul et al, 2015

---

> ### Author Response · Authors · 2025-06-29
> **Response to Reviewer C8RH**
>
> First, we would like to thank you for your review. We appreciate your comments and suggestions, which have helped us improve the manuscript. Changes are highlighted in blue, and below we provide detailed explanations for each requested revision and how we addressed them.
>
> =====Details about the relabeling and validation time and effort=====
>
> We acknowledge the importance of analyzing demonstration time and mental effort. Accordingly, we added a detailed discussion on demonstration times and user effort in Appendix A.6. This section presents a time-efficiency analysis following [1] of AIDA based on the experiments from Section 5.3, including measured timings for each demonstration modality. It also discusses three possible demonstration cost scenarios across different tasks relevant to the experiments in Section 5.2. We have also added a reference to this section in the limitations section (Section 6.2).
>
> =====Applicability of AIDA=====
>
> We agree that extending AIDA to higher-frequency, low-level control tasks is an interesting direction for future research. While the benefits of relabeling and validation within AIDA may diminish as query frequency increases, the SAG component remains relevant. For example, SAG can be employed to query a human teacher for supervision while robot execution can be slowed down, enabling a hybrid approach combining robot-gated queries with human-gated interventions. Moreover, it would indeed be promising to integrate robot-gated AIDA for higher-level affordance or skill learning with human-gated low-level learning via interventions, such as HG-DAgger [2]. This combined approach could yield a hierarchical control structure similar to SayCan [3], but with the advantage of on-policy training. In contrast to SayCan, where skills are pretrained through behavioral cloning and RL, and high-level affordances are learned offline with reinforcement learning informed by LLM predictions, our approach would facilitate simultaneous on-policy learning at both high and low levels, effectively gathering demonstrations at different control frequencies. We have included these as future research directions in the discussion section.
>
> =====Comparison with affordance-based/reinforcement learning methods=====
>
> We agree with the reviewer that a comparison to affordance learning methods, such as Where2Act [4] and Information-Driven Affordance Discovery [5], is relevant since AIDA also learns policies at the affordance level. We have therefore included a paragraph on affordance learning methods in Section 2 (Related Work) and clearly discuss the key differences between these approaches and AIDA. A primary difference is that methods like [4] and [5] focus on actively and efficiently exploring large affordance state-action spaces in simulators, where reward signals are readily available. In contrast, AIDA is explicitly designed to collect on-policy data aimed at optimizing performance during novice policy execution by learning from occasional teacher feedback. To illustrate the differing objectives, we implemented AIDA in a Where2Act task and added comparative experiments with Where2Act in Appendix A.12.
> Additionally, within reinforcement learning, the line of research known as Reinforcement Learning from Human Feedback (RLHF) [6] is closely related to our work. Following the reviewer’s recommendation, we have therefore expanded our related work discussion in Section 2 to include closely related RLHF literature, explicitly highlighting key distinctions between these methods and our approach.
>
> =====Performance plot for a sum of demonstrations=====
>
> As the discussion of this figure also relates to demonstration cost considerations, we have included this figure in Appendix A.6 together with an extensive discussion for different demonstration cost scenarios.
>
> =====A more detailed illustration of the method=====
>
> We thank the reviewer for this suggestion, and we have updated Figure 1 to provide more details on incoming and outgoing signals and to better connect the figure with the symbols and notations used throughout the text.
>
> References:
> [1] Johns, E. Back to reality for imitation learning. CoRL. 2022
> [2] Kelly, M., Sidrane, C., Driggs-Campbell, K., & Kochenderfer, M. J. HG-DAgger: Interactive imitation learning with human experts. ICRA. 2019
> [3] Brohan, A., Chebotar, Y., Finn, C., Hausman, K., Herzog, A., Ho, D., ... & Fu, C. K. Do as I can, not as I say: Grounding language in robotic affordances. CoRL. 2023.
> [4] Mo, K., Guibas, L. J., Mukadam, M., Gupta, A., & Tulsiani, S. Where2Act: From pixels to actions for articulated 3D objects. ICCV. 2021
> [5] Mazzaglia, P., Cohen, T., & Dijkman, D. Information-driven affordance discovery for efficient robotic manipulation. ICRA. 2024
> [6] Timo Kaufmann, Paul Weng, Viktor Bengs, and Eyke Hüllermeier. A survey of reinforcement learning from human feedback. TMLR, 2025

---

### Decision · Action_Editor_kAz5 · 2025-07-16

**Recommendation:** Accept as is

**Audience:**

Yes

**Audience Explanation:**

All reviewers believe that the paper is interesting to TMLR's audience, and I agree.

**Claims And Evidence:**

Yes

**Claims Explanation:**

At the conclusion of the review process all reviewers argue that the paper presents accurate, convincing, and clear evidence, and I agree. Reviewers praised the paper for its clear writing and explicitly stated claims, and believe that the empirical results are promising. Before the revision of the manuscript there were some concerns about discussing related work and having stronger baselines to compare to (that are not ablations of the methods). The authors have successfully addressed these concerns with an extended related work discussion and by adding two more baselines. All reviewers suggest accepting the paper as is, and I have nothing more to add.